# Closed and open structures of the eukaryotic magnesium channel Mrs2 reveal the auto-ligand-gating regulation mechanism

Ping Li [1] ✉, Shiyan Liu[2], Johan Wallerstein[3], Rhiza Lyne E. Villones[4], Peng Huang[1], Karin Lindkvist-Petersson [1], Gabriele Meloni [4], Kefeng Lu [2], Kristine Steen Jensen [3], Sara I. Liin [5] & Pontus Gourdon [1,6] ✉

The CorA/Mrs2 family of pentameric proteins are cardinal for the influx of $Mg^{2+}$ across cellular membranes, importing the cation to mitochondria in eukaryotes. Yet, the conducting and regulation mechanisms of permeation remain elusive, particularly for the eukaryotic Mrs2 members. Here, we report closed and open Mrs2 cryo-electron microscopy structures, accompanied by functional characterization. $Mg^{2+}$ flux is permitted by a narrow pore, gated by methionine and arginine residues in the closed state. Transition between the conformations is orchestrated by two pairs of conserved sensor-serving $Mg^{2+}$-binding sites in the mitochondrial matrix lumen, located in between monomers. At lower levels of $Mg^{2+}$, these ions are stripped, permitting an alternative, symmetrical shape, maintained by the RDLR motif that replaces one of the sensor site pairs in the open conformation. Thus, our findings collectively establish the molecular basis for selective $Mg^{2+}$ influx of Mrs2 and an auto-ligand-gating regulation mechanism.

In all living cells, $Mg^{2+}$ is the most abundant divalent cation[1]. Physiologically, $Mg^{2+}$ is an essential cofactor in a majority of cellular processes, involved in, for example, adenosine triphosphate (ATP) production[2–8]. Therefore, intracellular $Mg^{2+}$ homeostasis is tightly regulated by membrane protein $Mg^{2+}$ channels and transporters[2,9,10]. In human, dysregulation of the $Mg^{2+}$ levels is linked to a variety of diseases[11,12]. In eukaryotic cells, mitochondria represent the dominant compartment for production of energy in the form of ATP, a process requiring $Mg^{2+}$ as a cofactor, and the nucleotide also generally forms complexes with this cation in cells[13].

The mitochondrial inner membrane protein Mrs2 (mitochondrial RNA-splicing protein 2) is responsible for regulating the $Mg^{2+}$ concentrations of mitochondria in eukaryotes, providing controlled uptake of the ion[14–16]. Orchestration of the Mrs2-associated flux is critical, as upregulated expression of Mrs2 causes the $Mg^{2+}$ levels in mitochondria to elevate, which represents a hallmark of several types of cancers[17]. Conversely, deletion of Mrs2 eliminates $Mg^{2+}$ permeation into the matrix, resulting in abnormal mitochondrial function and cell death[15].

Mrs2 belongs to a superfamily of $Mg^{2+}$ channel-forming proteins known as CorA (because of mutants exhibiting $Co^{2+}$ resistance[18–20]) in prokaryotes and as Mrs2 in eukaryotes. The most well-studied members are of prokaryotic origin, which provide the cellular import of certain divalent ions such as $Mg^{2+}$ (refs. 18–21). Yet, CorA and Mrs2 are known to complement each other in eukaryotes and prokaryotes, respectively[22,23]. CorA/Mrs2 forms homopentamers with two transmembrane (TM) helices (TM1 and TM2) in each monomer, which are

[1]Department of Experimental Medical Science, Lund University, Lund, Sweden. [2]Department of Neurosurgery, State Key Laboratory of Biotherapy, West China Hospital, Sichuan University, Chengdu, China. [3]Division of Biophysical Chemistry, Center for Molecular Protein Science, Department of Chemistry, Lund University, Lund, Sweden. [4]Department of Chemistry and Biochemistry, The University of Texas at Dallas, Richardson, TX, USA. [5]Department of Biomedical and Clinical Sciences, Linköping University, Linköping, Sweden. [6]Department of Biomedical Sciences, Copenhagen University, Copenhagen, Denmark. ✉e-mail: ping.li@med.lu.se; pontus.gourdon@med.lu.se

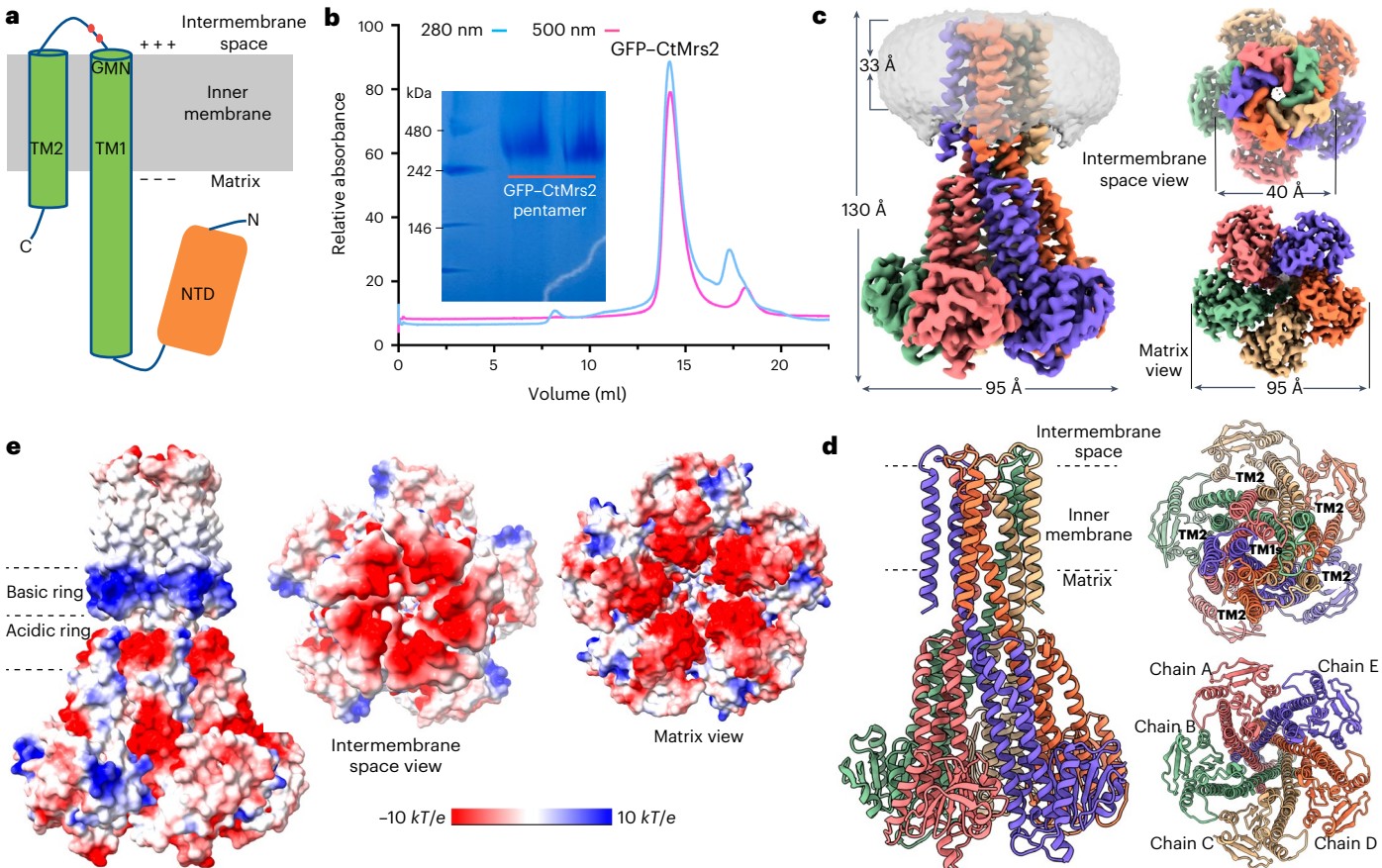

**Fig. 1 | Overall architecture of Mrs2 and the closed conformation. a**, Topology of one Mrs2 monomer, with the NTD located in the matrix and TM1 and TM2 in the C terminus. The conserved GMN selective filter motif is located at the end of TM1 and two acidic residues (red dots) in the loop connecting TM1 and TM2. **b**, SEC profile of detergent-solubilized GFP-fused CtMrs2 (performed at least five independent times) and associated Coomassie-stained native PAGE (performed twice independently), indicating that CtMrs2 forms homopentamers. **c**, The 2.7-Å overall resolution cryo-EM density of the closed CtMrs2 homopentamer, shown from the membrane plane, from the intermembrane space and from the matrix (C5 map). Separate monomers are colored blue, orange, wheat, green and pink and the nanodisc is colored gray. **d**, Cartoon representation of the final structure of the closed CtMrs2 homopentamer with the same views and colors as in **c**. **e**, Surface electrostatics of the closed CtMrs2 homopentamer, shown from the membrane plane, from the intermembrane space and from the matrix.

preceded by a soluble N-terminal domain (NTD) that regulates the Mg²⁺ passage (Fig. 1a)[24–27]. The protein family exposes only the loop between TM1 and TM2 to the surrounding environment but this stretch harbors a conserved selective filter GMN motif that is essential for permeation[25,28–31]. While it appears as if different residues in the pore establish restriction to orchestrate Mg²⁺ conductance in CorA and Mrs2 (refs. 24,25,30–33), respectively, it is still poorly understood how the protein family operates at the molecular level. Proposed opening models for CorA include an iris-like widening of the channel upon Mg²⁺ binding to the regulatory domain[26,34] and exchange of the pore-lining amino acids to hydrophilic residues through rotation of the stalk[25,27], respectively. Notably, such gating processes are dependent on major conformational changes between closed, symmetric and open, asymmetric states in the presence and absence of Mg²⁺, respectively[31,35–37].

Apart from studies of isolated domains[38], the available CorA/Mrs2 structural information is limited to nonconducting structures of prokaryotic TmCorA[24,30–33] from *Thermotoga maritima*, MjCorA[25] from *Methanocaldococcus jannaschii* and human Mrs2 (hMrs2)[39,40]. Moreover, low-resolution structural information is available for TmCorA, suggesting an opening through an asymmetric conformation. Consequently, several considerable research gaps remain in the field. Most notably, the molecular principles that govern permeation, gating and regulation remain enigmatic, particularly for Mrs2 proteins but also for CorA. This is further underscored by the available conflicting

opening models for CorA and the relatively low sequence homology between Mrs2 and CorA proteins and even among Mrs2 members.

Herein, we provide results that shed further light on the Mg²⁺ uptake and regulation of Mrs2.

## Results

### Functional characterization and overall architecture of Mrs2

Reduced growth on nonfermentable carbon sources represents a classical phenotype of eukaryotic cells lacking Mrs2 (ref. 38). We reproduced this observation through experiments using an in vivo assay based on wild-type (WT) and *MRS2*-knockout *Saccharomyces cerevisiae* strains (Extended Data Fig. 1a). To further dissect the function of the Mrs2 proteins, we selected a member from *Chaetomium thermophilum*, CtMrs2 (UniProt G0S186), which shares 30% and 43% sequence identity to hMrs2 and ScMrs2, respectively (Extended Data Fig. 2). First, we assessed the cellular localization in *S. cerevisiae* cells of a conservatively truncated CtMrs2 form fused to green fluorescent protein (GFP) to the N terminus. As observed from the bioimaging of live cells, the protein is targeted to internal compartments (Extended Data Fig. 1b), consistent with localization to mitochondria. Furthermore, the purified GFP-fused CtMrs2 protein sample displayed the expected homopentameric state following size-exclusion chromatography (SEC) and on blue native page (Fig. 1b). Thus, our data suggest that CtMrs2 serves as a Mg²⁺-conducting homopentameric channel resident in mitochondria like other Mrs2 members.

To further illuminate the structure and function of Mrs2 channels, the purified CtMrs2 sample was reconstituted into nanodiscs to facilitate single-particle cryo-electron microscopy (cryo-EM) (Methods). We first maintained $Mg^{2+}$ in all procedures during sample preparation. Cryo-EM micrographs of the sample showed well-distributed particles, with the corresponding two-dimensional (2D) classifications demonstrating features of a soluble and a TM domain, respectively (Extended Data Fig. 3). The final reconstructed map reached an average resolution of 3.1 Å in the absence of imposed symmetry (Table 1). Upon map inspection, we also obtained a complementary map determined at 2.7-Å overall resolution when *C5* symmetry was applied, in accordance with the established five-fold assembly of CorA/Mrs2 proteins[31]. The TM and soluble domains are overall well resolved in the *C5* map, both consistent with five intermixed monomers, enabling de novo building of the polypeptide chains (Fig. 1c,d, Extended Data Figs. 3 and 4 and Supplementary Fig. 1a). The CtMrs2 architecture exhibits a similar cone-shaped fold to the prokaryotic CorA proteins (root-mean-square deviation (r.m.s.d.) of 3.4 Å to Protein Data Bank (PDB) 3JCF; alignments were performed using secondary-structure matching in Coot throughout the manuscript)[24,25,30–33]. Moreover, the structure shows an even higher resemblance to the structure of hMrs2 (r.m.s.d. of 3.8 Å to PDB 8IP3) (Extended Data Fig. 5)[39–41]. Accordingly, the membrane-spanning domain comprises two helices from each monomer. TM1 (residues 416–441) helices establish an inner ring ~30 Å across the mitochondrial inner membrane, surrounding the pore that is vertical to the membrane. TM1 helices also form a funnel extending ~100 Å into the matrix (sometimes referred to as the stalk helix, residues 371–440), while TM2 (residues 451–483) helices form an outer ring, wrapping around the TM1 helices. Because the TM1 helices are twisted (approximately 20° compared to TM2 helices), the TM2 helices interact with two TM1 helices, one within the monomer and the other in the adjacent polypeptide. Conversely, the NTD harbors six antiparallel β-sheets (β1–β6) and eight helices (α1–α8) (Fig. 1c,d and Extended Data Fig. 1c,d). Helices α4 and α5 (also known as willow helices) are parallel to the stalk helix, forming an acidic ring adjacent to the membrane interface through a range of negatively charged residues (Fig. 1e and Extended Data Fig. 1e). Adjacent to this acidic region, a basic ring is formed by positively charged residues of TM1 (K422) and the C terminus. The structure also exhibits an electronegative entry mouth from the intermembrane space (Fig. 1e), which may have a role in attracting hydrated $Mg^{2+}$ from the surrounding environment. This region is established by the loop in between TM1 and TM2, which is possibly partly integrated in the mitochondrial inner membrane (Fig. 1c,d). Similarly, the inside of the funnel in the matrix is also highly negatively charged (Fig. 1e). These surface charge features are also preserved overall in hMrs2 (refs. 39–41).

## A closed configuration with multiple bound $Mg^{2+}$ ions

While the ion permeation pore overlays with that of the CorA proteins, it is substantially longer in CtMrs2, stretching ~70 Å across the mitochondrial inner membrane and remaining narrow a further ~5 Å into the matrix, limited by the five TM1 or stalk helices throughout (Fig. 2a,b and Extended Data Fig. 6). The channel starts at the negatively charged entrance in the intermembrane space and ends where the funnel commences to widen, stretching from N443 to R406, as also visualized using the MOLE online server[42] and HOLE software analysis[43] (Fig. 2a,b). This pore harbors four strong nonproteinaceous features in the cryo-EM density assigned as $Mg^{2+}$ ions, as supported by similar observations for certain of the sites in hMrs2, also in nonsymmetric *C1* maps[39,40] (Fig. 2c,d and Extended Data Fig. 7).

The first $Mg^{2+}$ is bound by the loops connecting TM1 and TM2, interacting with the conserved N443 (of the GMN motif), E449 and perhaps E450. This is relevant as it has been shown that this region is important for the $Mg^{2+}$ conductance of CorA and Mrs2 proteins[5,44]. Indeed, N443 and E449 are omnipresent within the CorA/Mrs2

**Table 1 | Cryo-EM data collection, refinement and validation statistics**

| | CtMrs2 closed (EMD-18256) (PDB 8Q8P) | CtMrs2 open (EMD-18257) (PDB 8Q8Q) |
|---|---|---|
| **Data collection and processing** | | |
| Magnification | 105,000 | 105,000 |
| Voltage (kV) | 300 | 300 |
| Electron exposure (e⁻ per Å²) | 50 | 49.958 |
| Defocus range (µm) | −0.6 to −2.2 | −0.6 to −2.0 |
| Pixel size (Å) | 0.8617 | 0.8566 |
| Symmetry imposed | *C5* | *C5* |
| Initial particle images (no.) | 663,423 | 332,383 |
| Final particle images (no.) | 222,578 | 116,609 |
| Map resolution (Å) | 2.7 | 3.2 |
| FSC threshold | 0.143 | 0.143 |
| Map resolution range (Å) | 2.57–6.56 | 3.21–7.99 |
| **Refinement** | | |
| Initial model used (PDB code) | AlphaFold model (AF-G0S186-F1) | 8Q8P |
| Model resolution (Å) | 2.9 | 3.4 |
| FSC threshold | 0.5 | 0.5 |
| Model resolution range (Å) | - | - |
| Map sharpening *B* factor (Å²) | −109 | −120 |
| **Model composition** | | |
| Nonhydrogen atoms | 12,579 | 12,718 |
| Protein residues | 1,585 | 1,575 |
| Ligands | MG: 24 | LOP: 5; MG: 3 |
| **B factors (Å²)** | | |
| Protein | 46.95 | 57.57 |
| Ligand | 41.09 | 42.36 |
| R.m.s.d. | | |
| Bond lengths (Å) | 0.002 | 0.003 |
| Bond angles (°) | 0.380 | 0.497 |
| **Validation** | | |
| MolProbity score | 1.11 | 1.28 |
| Clashscore | 3.15 | 2.94 |
| Poor rotamers (%) | 0 | 0 |
| Ramachandran plot | | |
| Favored (%) | 98.73 | 96.78 |
| Allowed (%) | 1.27 | 3.22 |
| Disallowed (%) | 0.00 | 0.00 |

MG, $Mg^{2+}$; LOP, lauryl oleyl phosphatidylethanolamine.

family and E450 is also conserved among Mrs2 (Extended Data Figs. 2 and 5a). We propose that E449 and E450 are responsible for the initial uptake of fully hydrated $Mg^{2+}$ (with a diameter of 9.5 Å)[1], as permitted by the 10.6-Å and 17.8-Å diameters of the pores at these residues (pore calculations were computed using the software HOLE throughout the manuscript), respectively, a hypothesis also supported by previous efforts on hMrs2 (ref. 5). This arrangement may prime hydrated $Mg^{2+}$ for the pore, thereby augmenting the local concentration before uptake, and contribute to establishing specificity. We coin this site U, as it likely is important for the uptake of partially dehydrated $Mg^{2+}$,

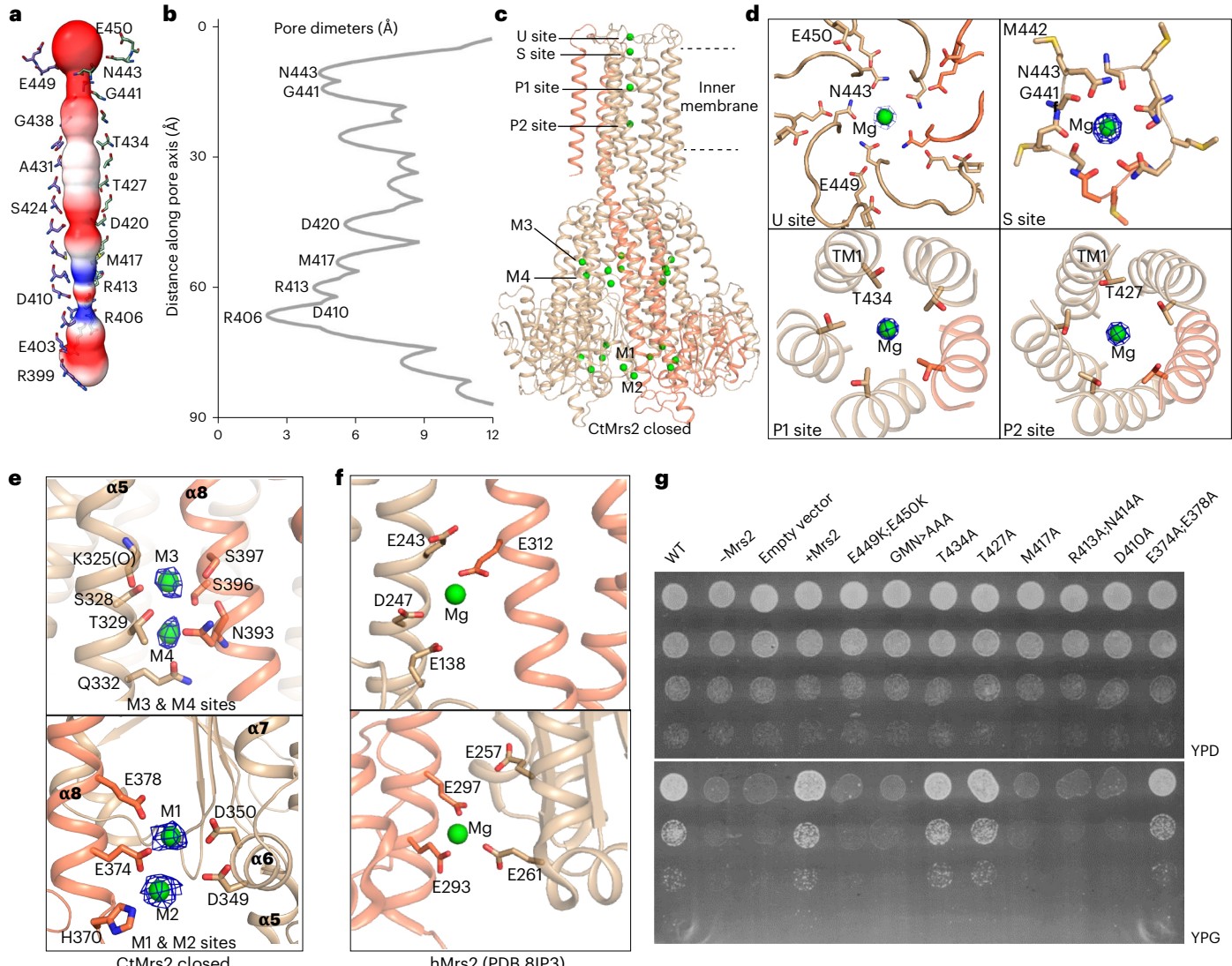

**Fig. 2 | The closed permeation pathway and Mg²⁺-binding sites.** One selected monomer is shown in orange throughout. **a**, MOLE software analysis of the conducting pore, shown as electrostatic surface, with residues lining the pathway shown as sticks. **b**, HOLE software calculation of pore diameter along the pore. **c**, Location of putative Mg²⁺ ions (shown in green) in the closed homopentamer. **d**, Close-up views with supporting cryo-EM density in the symmetry-applied *C5* map of the putative Mg²⁺-binding sites close to or in the conducting pathway. Site U is positioned next to the loop in between TM1 and TM2 and site S is positioned next to the GMN motif of the selectivity filter. Sites P1 and P2 are located in the TM domain. **e**, Close-up views of sites M1–M4 in the NTD of CtMrs2. **f**, Close-up views of the corresponding Mg²⁺-binding sites (shown in **e**) in hMrs2 (PDB 8IP3), equivalent to sites M1 (top) and M4 (bottom). **g**, Growth phenotypes on YPD and YPG (the latter requiring mitochondrial respiration) media of WT Mrs2 from *S. cerevisiae* (ScMrs2), the equivalent *MRS2*-knockout strain (−*MRS2*), and cells based on −*MRS2* with an empty vector or a vector containing different mutant forms of ScMrs2 (CtMrs2 numbering).

considering the 4.6-Å distance to N443. Directly associated, a second Mg²⁺ ion is coordinated to the main and side chains of GMN motif residues G441 and N443, respectively (Fig. 2c,d and Extended Data Fig. 7). The narrow width of the pore (~4.5-Å diameter at both G441 and N443) suggests that the ion is further dehydrated, as previously reported for this site of TmCorA[31–33] and hMrs2 (ref. 39); thus, Mg²⁺ is partially water-stripped during the delivery from site U. Further supporting a critical role of the second site, it is generally found in CorA and Mrs2 structures, including hMrs2 (we name this site S for specificity, as discussed later)[31–33,39].

Finally, two additional features of the cryo-EM density in the pore are assigned as Mg²⁺ ions, the first located at T434 (glycine in hMrs2) and the second positioned at the conserved T427 (Fig. 2c,d and Extended Data Fig. 7). We designate these pore sites P1 and P2; because of the narrow pore diameters of 5.3 and 7.4 Å, respectively, it is likely that the ions at these sites are only partially hydrated.

Three more constriction regions of the pore follow P2, marked by D420 (asparagine in hMrs2), the conserved M417–R413 pair that is positioned at the membrane interface to the matrix and the R406 (phenylalanine in hMrs2)–D410 (invariant) pair. Yet, we do not observe cryo-EM density for ions at these three constrictions. However, on the basis of the hMrs2 structures, the presence of Cl⁻ and Mg²⁺ was suggested at the equivalent of R413 and D410, respectively[39,40]. Nonetheless, the combination of invariant positively charged residues at R413 and a hydrophobic or positively charged lock for cations at R406, along with the small diameters at these constrictions (4.0 and 2.0 Å, respectively, representing the narrowest parts of the permeation pathway), indicates that the CtMrs2 structure is closed (Fig. 2b).

**Intermonomer Mg²⁺ binding to the soluble domain**

The cryo-EM density reveals the presence of four more putative Mg²⁺-binding sites per monomer, present as a 'glue' in between separate

polypeptides in the soluble fraction of CtMrs2. One pair of these sites (M1 and M2; M for the matrix of the mitochondria) are positioned at the tip of the protein (Fig. 2c,e and Extended Data Fig. 7). M1 is tightly coordinated by E374 and E378 of the stalk helix of one monomer and linked to D349 and D350 next to α5 of the adjacent monomer (Fig. 2c,e and Extended Data Fig. 7). Notably, this site is essentially conserved in hMrs2 as E293, E297 and E261 (Fig. 2f). Similarly, M2 interacts with H370 and the conserved E374 and loosely binds to the conserved D349 of the neighboring protein chain (Fig. 2c,e and Extended Data Fig. 7). M2 may also exist in hMrs2 as E262 and E293 or E261 and E263 but this is not supported by the hMrs2 structures[39–41]. We conclude that the approximate position of M1 likely is conserved, although the binding amino acids vary somewhat among Mrs2 members. Divergent to the $Mg^{2+}$ sites at the funnel tip in CorA proteins, M1 and M2 are located inside the negatively charged funnel, which may imply a somewhat different role of these sites (or rather M1) in Mrs2 proteins than the corresponding in CorA proteins (Extended Data Fig. 6c).

The second pair of putative $Mg^{2+}$-binding sites (here denoted M3 and M4) are located in between the stalk helix of one monomer and helix α5 of the adjacent monomer (Fig. 2c,e and Extended Data Fig. 7). M3 and M4 are positioned deeper into the funnel than M1 and M2, closer to the pore end. M3 is coordinated by S396 and S397 of the stalk helix and K325, S328 and T329 (through the side chains for all except the lysine) in the α5 helix of the neighboring monomer. Conversely, the directly associated M4 interacts with N393 and S396 of the stalk helix and S328, T329 and Q332 of helix α5 of the adjacent protein chain (Fig. 2c,e and Extended Data Fig. 7). Interestingly, T329 and N393 are conserved as glutamates in many Mrs2 members including hMrs2. Moreover, despite being positioned somewhat more peripherally, M4 is maintained in hMrs2, bound by the conserved E312 of the stalk helix and E138, E243 and D247 of helix α5 (Fig. 2f)[39]. Similarly, sites M3 and M4 are also conserved in certain but not all CorA members, such as MjCorA (PDB 4EV6)[25] (Extended Data Fig. 5d). Thus, in addition to site M1, it is likely that the approximate location of site M4 is maintained across Mrs2 proteins (subtle differences observed even among hMrs2 structures, sometimes involving T246)[40] and that M4 is also present in many CorA members. It is possible that the highly electronegative environment of the funnel serves to ensure that these cation sites are occupied when the $Mg^{2+}$ levels in the matrix are elevated (Fig. 1e).

### On the functional role of the $Mg^{2+}$-binding residues

To further investigate $Mg^{2+}$ binding, conductance and regulation, we exploited the previously mentioned *S. cerevisiae*-based assay, assessing the WT and mutant forms of ScMrs2 with maintained expression profiles (Extended Data Figs. 5a and 8 and Supplementary Fig. 1b). Supporting a role in ion uptake and permeation for site U, growth on a nonfermentable carbon source was impaired with the E449K;E450K double substitution (Fig. 2g). Similarly, the G441A;M442A;N443A mutant exhibited reduced cell proliferation, which is consistent with previous studies demonstrating that interruptions of the GMN motif impair the protein function[23,44–46]. Surprisingly, substitutions targeting sites P1 and P2 (T434A and T427A, respectively) left the cells essentially unaffected. Our interpretation of this observation is that certain parts of the ion conductance pathway are somewhat insensitive to amino acid changes, as also supported by the fact that T434 is replaced by a glycine in hMrs2 (Extended Data Figs. 2 and 5a). Conversely, the cell growth was abolished when selected, putative, gating residues of the pore were substituted (M417A, N412A;R413A and D410A), indicating a crucial role of these amino acids for protein function (Fig. 2g). However, substitution of two of the residues contributing to the M1 and M2 sites (E374A;E378A) left the cell proliferation unaffected. This can be interpreted as the latter form having little consequence on protein function, as discussed later.

### The open state

While not evident from the previously available structures of CorA/Mrs2, Mrs2 must open for $Mg^{2+}$ influx when the levels of the cation are low. Consequently, to further illuminate the $Mg^{2+}$ conductance mechanism, we prepared a new CtMrs2 sample using a mild isolation strategy with the inclusion of EDTA from cell lysis and until final usage. Two cryo-EM maps, with and without applied *C5* symmetry, were determined at an overall resolution of 3.2 and 3.5 Å, respectively (Extended Data Fig. 9 and Methods). However, the two maps superpose well, yielding a single structure (Fig. 3a, Extended Data Fig. 10 and Table 1).

The EDTA-induced structure preserves the homopentamer form (Fig. 3a and Extended Data Fig. 10). Indeed, the TM domain including the pore is well maintained overall and the intermembrane space remains negatively charged, presumably facilitating $Mg^{2+}$ uptake (Fig. 3b–d). However, as partly achieved by rotation of these side chains away from the center (discussed below), the constriction rings of the pore marked by D420, M417–R413 and R406 in the closed structure display a striking enlargement of about 1, 5 and 13 Å (Fig. 3d–f). As a consequence, site S at the GMN motif is the narrowest point of the pore (4.0–4.7 Å in diameter) in the open structure. Lastly, the funnel on the matrix side of the membrane is also widened. Thus, we conclude that this structure represents a conducting conformation, allowing the influx of partially hydrated $Mg^{2+}$ to the matrix (Fig. 3c,g,h).

Few intermonomer contacts remain in the funnel of the open structure. However, while the conserved R314 has a peripheral orientation in the closed configuration, it serves as a wedge in between two adjacent stalk helices in the open conformation, through interaction with the invariant D410 and N414 of the neighboring monomer, thus directly influencing the matrix-facing end of the conducting pathway (Fig. 3a,i). Furthermore, toward the funnel tip, residues R200 and R203 of the conserved RDLR motif of the α1–α3 subdomain interact with E346, D348, E374 and E378 of the stalk helix of the adjacent monomer (side chain interactions except for E346), residues that assist in establishing the M1 and M2 sites in the closed structure (Fig. 3a,i). It is, thus, tempting to speculate that sites M1 and M4 (and sites M2 and M3 in CtMrs2) stabilize the closed state, while the RDLR site serves the same purpose in the open conformation. However, the RDLR motif is only present in the eukaryotic Mrs2 proteins, implying that this interaction is specific for the eukaryotic members.

Interestingly, while we expected a $Mg^{2+}$-depleted structure, we observed cryo-EM density at the S, P1 and P2 sites, which we tentatively assigned as $Mg^{2+}$ despite the EDTA treatment of the sample (the P2 site is not as distinct as the other sites in the map calculated without imposed symmetry) (Fig. 3j). To confirm whether $Mg^{2+}$ was still present in the presence of chelator, we determined $Mg^{2+}$ binding stoichiometries to CtMrs2 using inductively coupled plasma mass spectrometry (ICP-MS). The results show that $Mg^{2+}$ was indeed present in the sample subjected to EDTA (Fig. 3k). Surprisingly, we detected a similar $Mg^{2+}$-to-protein ratio for all samples ($Mg^{2+}$-treated sample, EDTA-treated sample and two separate double mutants targeting sites M3 and M4, respectively), equivalent to ~2 $Mg^{2+}$ ions per CtMrs2 pentamer as observed in the open structure (Fig. 3k). This indicates that CtMrs2 adopts an open conformation in the absence of $Mg^{2+}$, as we had to remove $Mg^{2+}$ in the last step of purification for the ICP-MS analyses. Considering that site S is formed by the hallmark GMN motif of the CorA/Mrs2 proteins, it is conceivable that its presence in the open configuration relates to an important role in establishing $Mg^{2+}$ specificity and perhaps also in preventing the (back)flow of $Mg^{2+}$, other ions or even water molecules from the matrix.

### Conformational changes between the closed and open states

In comparison to the TM domain, dramatic differences occur in the soluble domains between the closed and open structures. Overall, as seen from the mitochondrial matrix side, the soluble domains rotate counterclockwise and widen from the closed to the open structure,

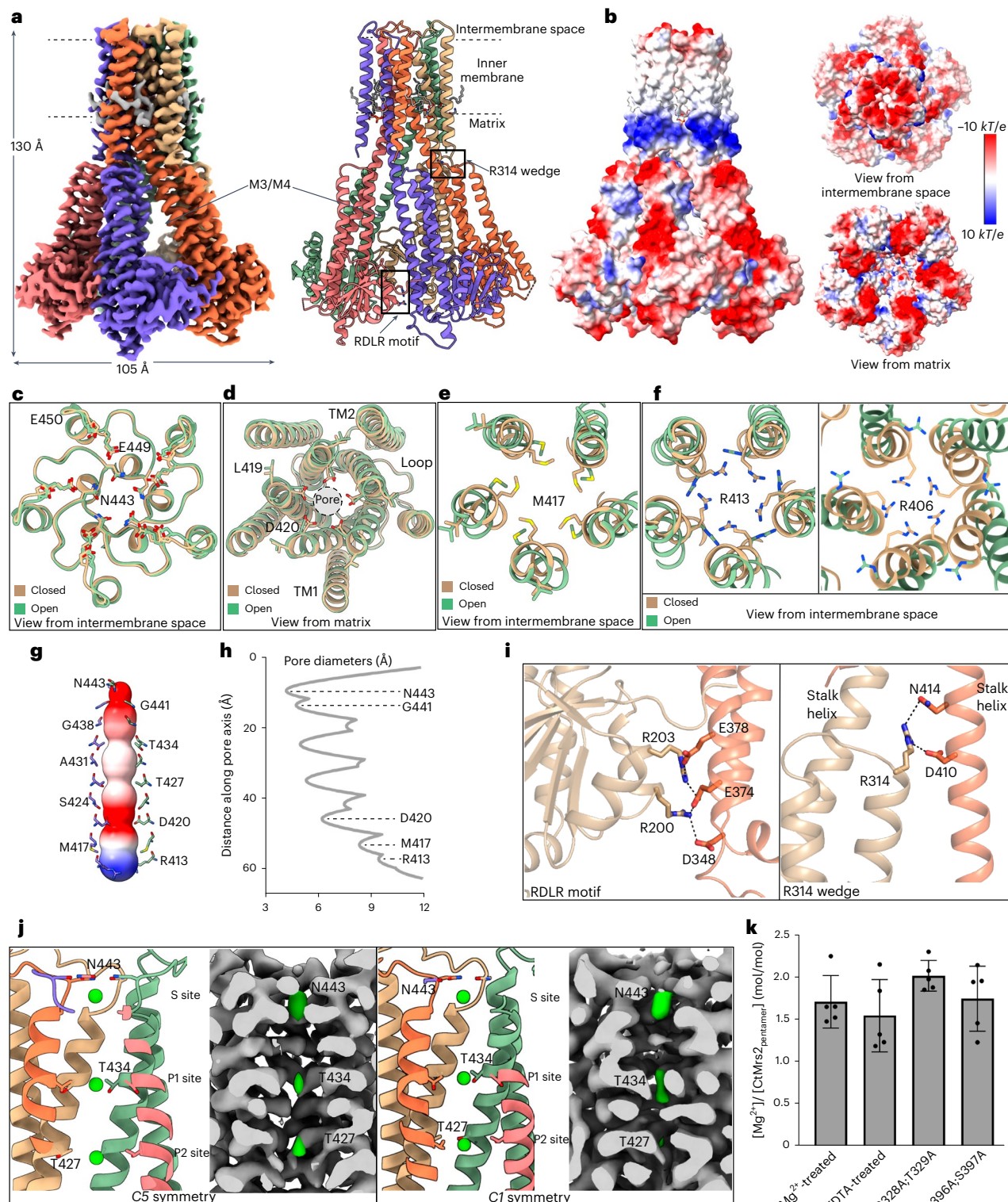

**Fig. 3 | The open state of Mrs2.** The cryo-EM density panels refer to the *C5* map throughout. **a**, Left, the 3.2-Å overall resolution cryo-EM density of the CtMrs2 homopentamer open state. Right, cartoon representation of the corresponding structure. The inset shows the feature assigned as cardiolipin, with the equivalent cryo-EM density in gray. **b**, Surface electrostatics of the open structure shown from the membrane plane, from the intermembrane space and from the matrix. **c**–**f**, Structural comparisons of the open and closed CtMrs2 structures along the ion conductance pore at E449-E450 (**c**; view from the intermembrane space), D420 (**d**; view from the matrix), M417 (**e**; view from the intermembrane space) and R413 and R406 (**f**; view from the intermembrane space). **g**,**h**, MOLE and HOLE software analyses of the pore of the open structure, shown as the electrostatic surface, with residues lining the pathway shown as sticks (**g**) and with the diameter of the pore along the pathway (**h**). **i**, Close-up views of the RDLR motif and the R314 wedge in the open structure. **j**, Putative $Mg^{2+}$-binding sites at the GMN motif selectivity filter (S site) and at the T427 (P1) and T434 (P2) rings observed in the cryo-EM maps calculated with five-fold symmetry (left) and without symmetry (right). **k**, $Mg^{2+}$-binding stoichiometry in various CtMrs2 forms under different conditions, as determined by ICP-MS (stoichiometries refer to $Mg^{2+}$ per CtMrs2 pentamer). Data points represent the means of five independent measurements, each from one purified sample, and error bars indicate the s.d.

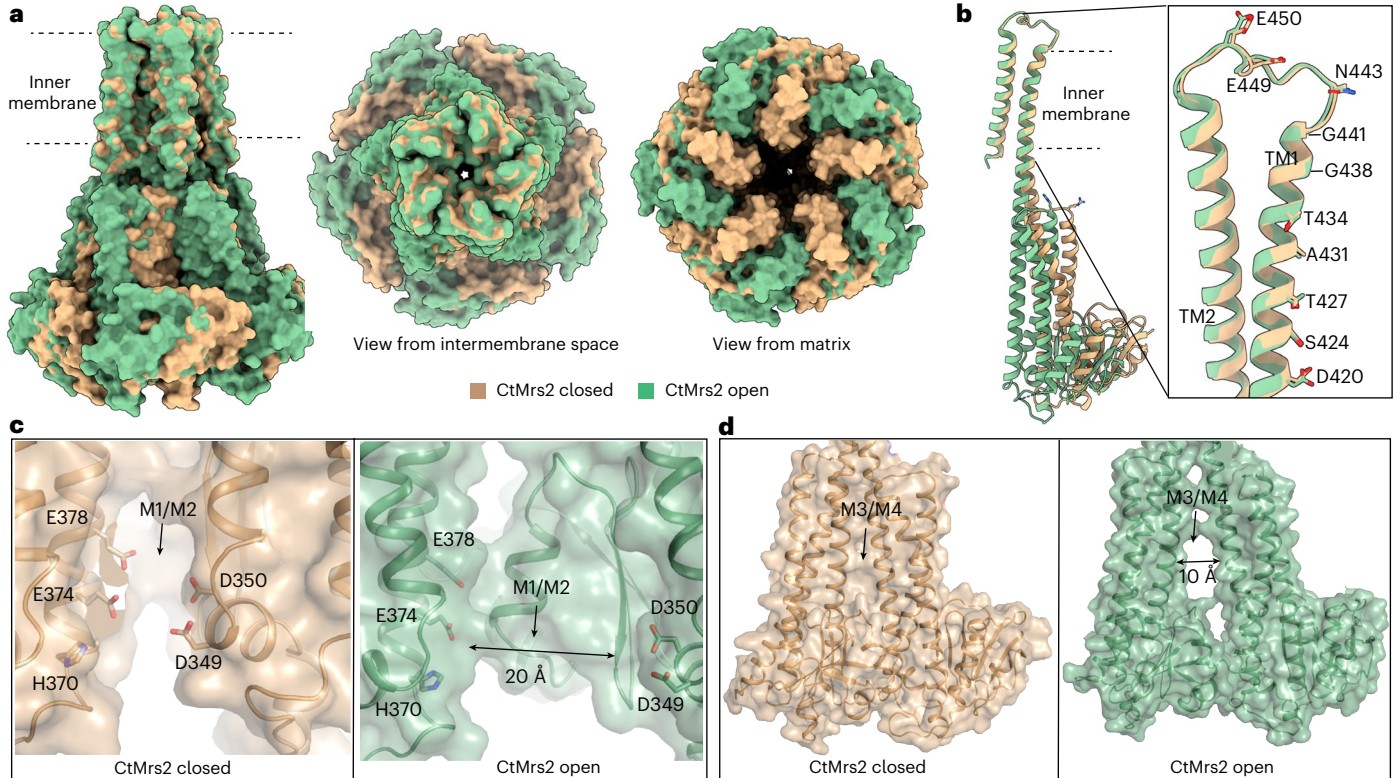

**Fig. 4 | Conformational changes between the open and the closed states.** **a,b**, Alignments of the TM domains of the open and closed CtMrs2 structures shown from the membrane plane, from the intermembrane space and from the matrix (**a**) and of a single monomer (**b**). Residues lining the pore are shown as sticks in **b**. The intermembrane space loop and the TM domain (formed by TM1 and TM2) are highly similar between the two states, whereas the NTD is rotated, providing a wider funnel in the open structure. Nonetheless, the five-fold symmetry is maintained in both configurations. **c,d**, Conformational changes at the M1 and M2 (**c**) and M3 and M4 (**d**) $Mg^{2+}$ ion-binding sites between the closed (left) and the open (right) states, with residues involved in ion binding shown as sticks.

with accentuating changes from the start (linked to the pore) to the end (at the tip) of the funnel (Fig. 4a,b).

Remarkably, among the most displaced structural features are the stalk and α5 helices of adjacent monomers, which are well separated in the open Mrs2 setting, thus eliminating the M1–M4 sites. Specifically, the residues forming the M1 and M2 sites are separated by more than 20 Å in the open structure (Fig. 4c). Similarly, the amino acids that bind to sites M3 and M4 are displaced by approximately 10 Å (Fig. 4d). The consequence is that the stalk helix is straightened and shifted outward in the open structure, thereby being mainly responsible for the widening of the D420, M417–R413 and R406 constrictions of the pore. Moreover, the stalk helix preceding the soluble domain (the α1–α3 subdomain) within the monomer is brought along from a funnel-lining to a more peripheral position. Notably, this greatly reduces contacts between monomers in the funnel in the open configuration, with the RDLR arginines essentially replacing the M1 site in the closed structure.

## The M1–M4 $Mg^{2+}$ site regions control the shape and conductance of Mrs2

Considering the dramatic rearrangements of the M1–M4 ion-binding regions between the two configurations, we aimed to further refine the roles of these sites. First, we used an *Escherichia coli*-based $Ni^{2+}$ sensitivity assay, which has been exploited, for example, for studies of hMrs2 (ref. [40]), assessing Mrs2-faciliated increased uptake of the metal. As expected, *E. coli* cells expressing CtMrs2 showed increased sensitivity toward $Ni^{2+}$ compared to control cells and the toxicity could be prevented through the N443A substitution that interferes with the GMN selectivity filter (Fig. 5a). Conversely, disruption of the M1 and M2 (E374A;E378A) or the M3 and M4 (S328A;T329A and S396A;S397A)

sites increases cell toxicity (the same mutants were applied in all below-mentioned assays). This suggests that such forms correlate with open CtMrs2. Instead, an intention to stabilize the M1 and M2 sites (E374R, mimicking bound $Mg^{2+}$) resulted in decreased $Ni^{2+}$ sensitivity, congruent with closed Mrs2. Interestingly, supplementation of $Mg^{2+}$ partially rescued all protein forms, indicating that CtMrs2 has higher affinity for $Mg^{2+}$ than for $Ni^{2+}$ and further illustrating that the M1–M4 $Mg^{2+}$-binding sites have an important mechanistic role (Fig. 5a,b).

To further dissect the conductance, we performed electrophysiological recordings of CtMrs2 and mutants, which were expressed in *Xenopus* oocytes and perfused extracellularly with $Mg^{2+}$ solution, guided by previous studies[41,46] (Fig. 5c,d and Supplementary Fig. 2). The results show that WT displayed clear inward currents that peaked within seconds of the supplementation of $Mg^{2+}$ and decayed slowly, reaching ~40% of the peak current by the end of the 2-min period that $Mg^{2+}$ was supplied. In contrast, no current was observed for the N443A form or in water-injected oocytes (control). Moreover, the inward currents were rapidly abolished upon return to a $Mg^{2+}$-free extracellular solution and were significantly reduced in the presence of the established inhibitor cobalt hexammine, all supporting that the CtMrs2 protein mediates the $Mg^{2+}$ influx. In agreement with the $Ni^{2+}$ sensitivity assay, significantly larger inward currents were detected for the ion-binding interfering forms (M1–M2 and M3–M4), which were even more pronounced for the M3–M4 mutants, in accordance with these amino acid changes favoring the open state. In comparison, no current was detected for the E374R mutant, supporting that this form prefers the closed conformation. Conversely, the decay of the flux during the 2-min $Mg^{2+}$ exposure was similar for WT and for the M3–M4 alanine substitutions but the conductance was better maintained for the M1–M2 alanine mutant.

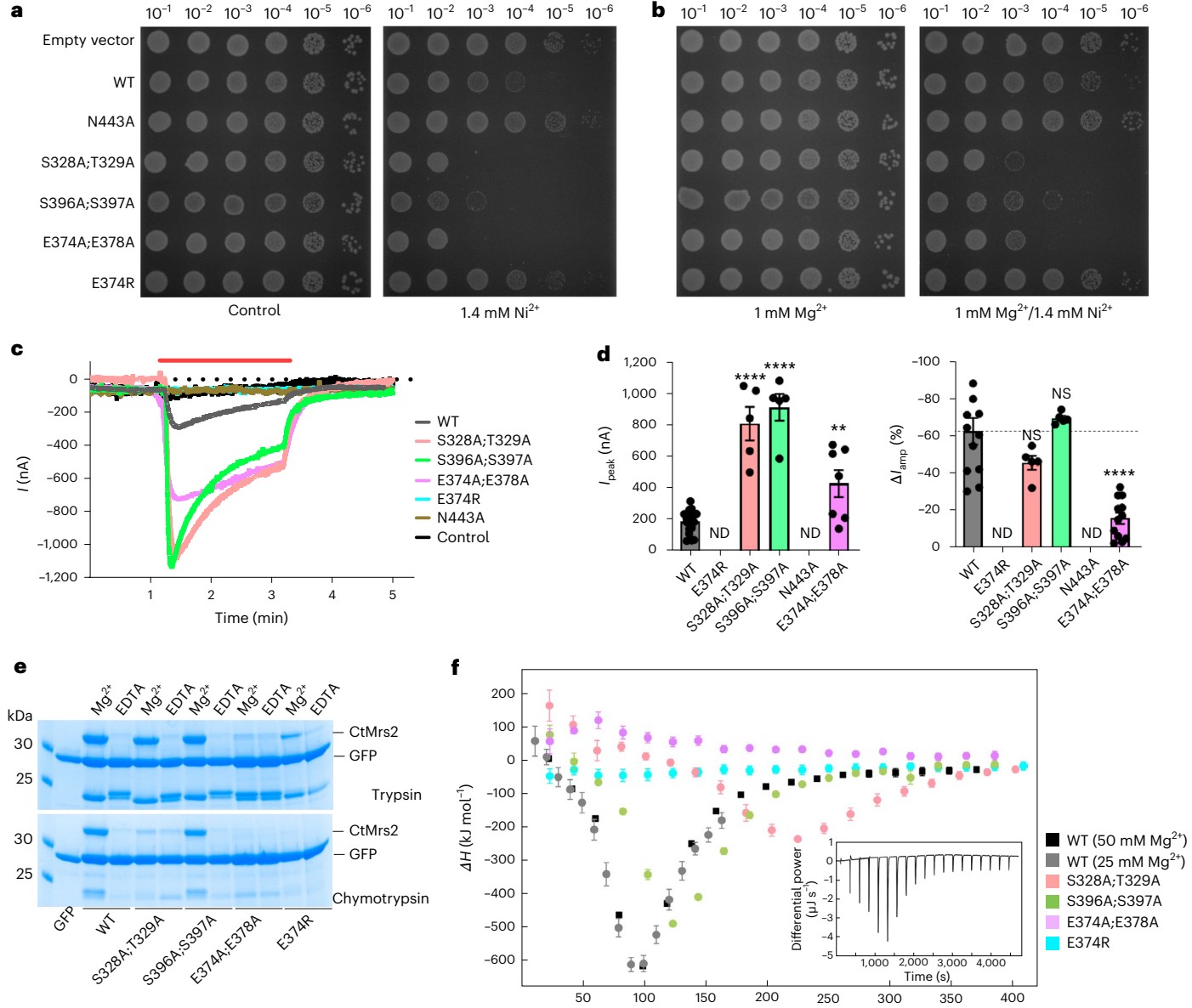

**Fig. 5 | The roles of the M1–M4 Mg²⁺-binding sites. a,b,** Comparison of the growth of *E. coli* with different CtMrs2 forms at different ion concentrations. The empty vector represents a control without Mrs2. **c,** Overlay of representative Mg²⁺ currents recorded in oocytes expressing WT and mutants. The red bar indicates the application period of Mg²⁺-containing recording solution. The control denotes water-injected oocytes. **d,** Summary of the recorded currents with peak current amplitudes following Mg²⁺ perfusion ($I_{peak}$; left) and spontaneous current decay during Mg²⁺ perfusion ($\Delta I_{amp}$ (%); right). Data shown as the mean ± s.e.m. ($n = 5$–19). Statistical analysis was conducted using a one-way ANOVA followed by Dunnett's multiple-comparisons test in comparison to WT. NS, nonsignificant ($P > 0.05$; 0.16 for S328A-T329A and 0.79 for S396A-S397A); **$P = 0.0036$ and ****$P < 0.0001$. Further data and details are provided in Supplementary Fig. 2 and the Methods. ND, not determined (because of no detectable currents). **e,** Limited proteolysis assay (performed

twice independently) using purified CtMrs2 forms using two separate proteases. The bands above 30 kDa represent CtMrs2, while the bands between 25 and 30 kDa are GFP, as confirmed by mass spectrometry. **f,** Binding isotherms for Mg²⁺ binding to WT CtMrs2 (at two stock concentrations) and mutant forms showing the heat of injection as a function of the molar ratio between Mg²⁺ and CtMrs2 monomer. For all mutants, 50 mM Mg²⁺ was used. For the mutants, the presented values are the means of two independent ITC titrations. Error bars show one s.d. and are estimated from the baseline uncertainties provided by NITPIC as previously described[54] and from the uncertainty between injection from the two datasets, where *i* refers to the injection number (1–19). The inset graph shows the raw thermogram, before integration by NITPIC, for the titration of 50 mM Mg²⁺ into WT. The complete ITC data are presented in Supplementary Fig. 4 (Methods).

Next, we set up a limited proteolysis assay using purified protein. All the purified CtMrs2 forms maintained the pentameric assembly as assessed by SEC and all samples were susceptible to proteases (trypsin or chymotrypsin) at low Mg²⁺ concentrations (Fig. 5e and Supplementary Fig. 3). However, at elevated Mg²⁺ levels, WT and the two M3–M4 site mutants preserved protease resistance (less so for the S328A;T329A form). Conversely, the two M1–M2

substitutions (E374A;E378A and E374R) displayed almost complete susceptibility to protease degradation in the presence of Mg²⁺ (some uncleaved CtMrs2 left with trypsin for E374R). This indicates that the open state is more vulnerable for cleavage and that the two M3–M4 mutant forms can shift from the open to the closed configuration when Mg²⁺ is supplemented, whereas the M1–M2 alanine substitution may not.

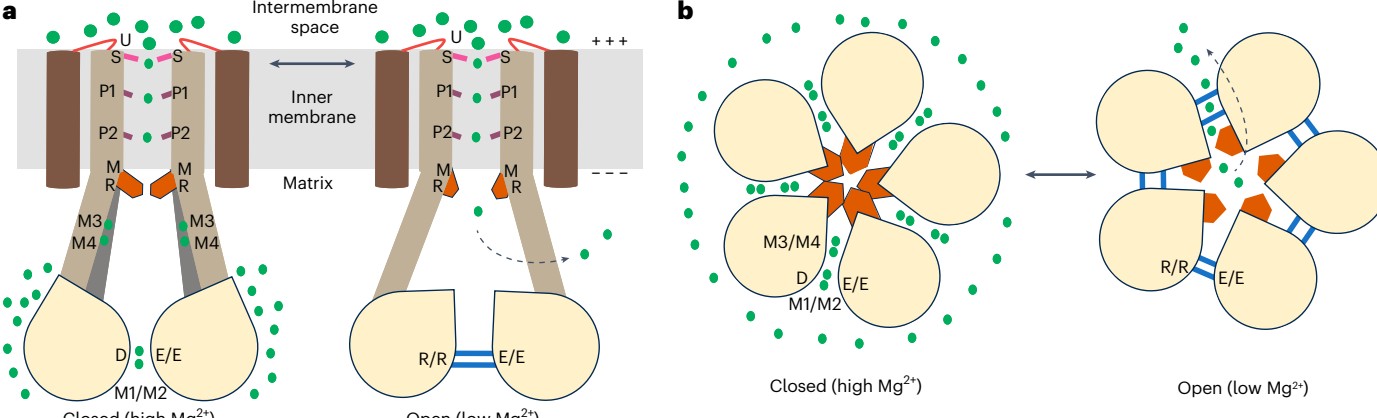

**Fig. 6 | Proposed Mg²⁺ autoregulation gating mechanism of Mrs2.**
**a,b**, Schematic model as viewed from the mitochondrial inner membrane (**a**; with two monomers for simplicity) and from the matrix (**b**). In the closed state, Mg²⁺ ions bind between the NTDs of separate monomers (positions M1 and M4 or M1–M4 in CtMrs2), thereby assisting in stabilizing the symmetrical shape, and to distinct positions of the pore (U, S, P1 and P2). In the open configuration, Mg²⁺ is only present at some of the pore sites (S, P1 and P2) and the NTD is instead maintained in an alternative symmetrical assembly by the RDLR motif and residues that previously formed the M1 and M2 sites (blue). At elevated Mg²⁺ levels in the mitochondrial matrix, Mrs2 is closed. Acidic rings formed by the loops in the intermembrane space (shown in red) attract fully hydrated Mg²⁺ (large green circles), which can be transferred as partially hydrated Mg²⁺ (small green circles) to the asparagine ring of the GMN motif selectivity filter (pink) and then to the P1 and P2 sites (purple) of the pore in the TM domain. However, flux is not permitted as the pore-gating methionine and arginine rings (brown polygons), located approximately at the membrane interface to the matrix, are closed. At low Mg²⁺ concentrations in the matrix, the open structure is present. Removal of the ions from M1 and M4 (and from M1–M2 and M3–M4 in CtMrs2) of the closed structure triggers a shift and rotation of the stalk helix (light brown), resulting in opening of the pore gate, which permits Mg²⁺ influx into the matrix. It is possible that site S also prevents backflow to the intermembrane space. The closed state is reobtained through Mg²⁺ destabilization of the RDLR motif interaction and through Mg²⁺ bridging of the separated residues of the M3–M4 site.

Lastly, we conducted isothermal titration calorimetry (ITC) measurements to detect Mg²⁺ binding to purified CtMrs2 (Fig. 5f and Supplementary Fig. 4). The binding isotherms were recorded by titration of Mg²⁺ into the protein, starting with conditions under which an open protein conformation is expected. The binding isotherms of WT were approximately biphasic, reflecting multiple binding events upon the addition of Mg²⁺, as seen for other metal-binding proteins[47]. The first observed binding event was endothermic, while the second binding event was exothermic (Fig. 5f, inset, and Supplementary Fig. 4). However, both phases were dominated by the large exothermic contribution (negative peaks) of the second binding event. These findings are consistent with a sequential binding model with high positive cooperativity between two sets of binding sites[48,49]; the first binding event is endothermic and has the lowest affinity for the ligand, whereas the second binding event is exothermic and has the highest ligand affinity. We interpret the endothermic binding event to originate from Mg²⁺ binding tightly coupled to conformational changes from the open to the closed state. This structural shift then renders Mrs2 in a Mg²⁺-binding competent state where the ions are bound through an exothermic binding event to the M3 and M4 sites. The later reaction is associated with a large negative enthalpy change ($\Delta H$) that dominates the ITC signal. Detailed quantitative analysis of the binding isotherms remains difficult because of the complex binding model originating from multiple unique binding sites and the conformation sensitivity of the system, together with the relative sparse amount of data points in the binding isotherms[50–53].

To further confirm this interpretation, we measured the equivalent Mg²⁺ binding to the abovementioned CtMrs2 M1–M2 and M3–M4 site variants. Notably, the M1–M2 site forms displayed considerable different binding isotherms compared to WT and the M3–M4 mutants. The lack of signal for the E374R substitution supports the notion that the M1–M2 binding site in the E374R variant is occupied by the arginine sidechain. However, the E374R variant, despite favoring the closed state, appeared to also leave the M3–M4 sites in a binding-incompetent state. Similarly, binding of Mg²⁺ to the E374A;E378A variant was also significantly altered compared to the WT, presumably preventing the formation of the closed state. Conversely, disruption of the M3 and M4 sites (S328A;T329A and S396A;S397A, respectively) still permitted Mg²⁺ binding. The effect on the binding isotherms was a shift of the minimum to higher molar ratios (more Mg²⁺ required) and a decrease in the binding enthalpy (Fig. 5f). These effects are again consistent with a sequential binding model with high cooperativity between two sets of sites (M1–M2 and M3–M4), where Mg²⁺ binding to sites M3 and M4 only takes place after Mg²⁺ binding to sites M1 and M2 and where the introduced substitutions weaken the affinity of sites M3 and M4 to Mg²⁺. The observed change in signal from the M3–M4 variants (decrease in the binding enthalpies, $\Delta H$) is expected when the affinity for the ligand is lowered because of the substitution.

Collectively, this functional characterization points toward the M1–M2 site region being decisive for determining the CtMrs2 conformation, providing a closed state when Mg²⁺ is bound (as mimicked by E374R; an exception is discussed below) and an open configuration when the ions are not present (ion binding prevented by the M1–M2 alanine substitutions), as observed in our structures. The functional assays identify that a relatively high Mg²⁺ concentration (in the micromolar range) induce the conformational changes. The role of the M3 and M4 sites appears to be less dramatic; however, on the basis of the abovementioned assays, they are clearly important at least for the regulatory function of CtMrs2.

## Discussion

Herein, we propose a Mg²⁺ auto-ligand-regulated permeation mechanism (Fig. 6). Fully hydrated Mg²⁺ from the intermembrane space is attracted and concentrated by two layers of negatively charged residues (E449 and E450) in the pentamer. Furthermore, E449 forms the inner ring and N443 may be involved in Mg²⁺ dehydration ion at site U. Next, the ion is transferred to site S, which represents the narrowest point of the open channel where specificity is established by the GMN motif, as supported by inactivating substitutions (Figs. 2g and 5). Nevertheless, the site is continuously present in the open and closed structures; thus, a role in also preventing (back)flow cannot be excluded. The passage of partially hydrated Mg²⁺ is then facilitated by hydrophilic main and/or

side chains, including the conserved T427. Next, ion exit to the matrix is orchestrated by gating residues (in particular, R413 and M417), which in turn are controlled by the soluble domains.

At elevated $Mg^{2+}$ concentrations in the matrix, Mrs2 is stabilized in a closed configuration through $Mg^{2+}$ that bridges the monomers, two of which (sites M1 and M4) are conserved in Mrs2, including hMrs2. Conversely, under $Mg^{2+}$-depleted conditions, the two arginines of the RDLR motif and R314 serve to principally replace the M1–M2 and M3–M4 sites as a glue between monomers This stabilizes an alternative configuration of monomer–monomer interactions as observed in the open state and yet maintains the five-fold symmetry. Consequently, it is appealing to suggest that reduced levels of $Mg^{2+}$ in the matrix strip the ions from the M1 and M4 sites (or M1–M4 in in CtMrs2), which is followed by the open configuration in the absence of $Mg^{2+}$, in which the monomers are more separated, a transition that is unlikely to occur before the $Mg^{2+}$ at M1 (or M1 and M2 in CtMrs2) has been liberated.

How is then the open state closed when the $Mg^{2+}$ levels rise? We propose that $Mg^{2+}$ is also key for this transition, disturbing the RDLR interaction, reforming the M1 site (M1 and M2 sites in CtMrs2) and eventually bridging the separated M4 site (M3 and M4 sites in CtMrs2) as a fully hydrated ion, thereby also displacing R314. Such a mechanism likely depends on the M1 site (M1 and M2 in CtMrs2) and the RDLR site as on and off sensors, serving as critical motifs to stabilize the two different configurations, thereby dictating the open-to-closed shift and vice versa. Conversely, the M4 site (M3 and M4 sites in CtMrs2) likely has more of a regulatory role because of increased separation in the open structure. Such a concept would also be consistent with a possible nonsymmetrical open state of TmCorA[31]. CorA proteins lack sites M1 and M2 but have similar sites that are more peripheral and some CorA proteins also have an equivalent of M4; together, these sites stabilize the symmetrical closed state. Conversely, the RDLR motif is absent in CorA; hence, the open state cannot be stabilized in the symmetrical shape observed here, which may lead to a collapse of the homopentamer.

Thus, our findings collectively shed light on the molecular architecture that permits ion uptake, permeation, gating and regulation of eukaryotic Mrs2 proteins, results that unify decades of observations on the CorA/Mrs2 family of proteins and that also resonate with other $Mg^{2+}$ transporter and channel protein families (Supplementary Discussion).

## Online content

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

## Methods

### Gene cloning

Genomic DNA from *C. thermophilum* (strain DSM1495) was used as a template to amplify full-length CtMrs2 using primers 1 and 2 and inserted into the pET-22b vector using NEBbulder HiFi DNA Assembly Master mix, generating construct pET-22b-CtMrs2 introns. This form was used to remove the two introns using primers 3, 4, 5 and 6 by overlapping PCR, yielding the pET-22b-CtMrs2 construct. To enhance protein expression and purification, the gene fragment coding for residues 106–539 was subcloned into the pEMBLyex4 vector using primers 7 and 8 with a 10xHis-GFP-G4S-TEV (tobacco etch virus) tag fused to the N terminus of CtMrs2$_{106-539}$, yielding the pEMBLyex4-His10 -GFP-G4S-TEV-CtMrs2$_{106-539}$ construct. All mutations were generated using this form as the WT. Primers are listed in Supplementary Table 1. All constructs were confirmed by sequencing.

### Protein production

The PAP1500 *S. cerevisiae* strain was used for protein production[55]. The plasmid pEMBLyex4-His10-GFP-G4S-TEV-CtMrs2$_{106-539}$ was transformed into the *S. cerevisiae* strain using the lithium acetate single-stranded carrier DNA–PEG method[56], plated on SD medium supplemented with 15 g L$^{-1}$ agar and then incubated at 30 °C for 3 days. Next, single colonies were inoculated in 5 ml of SD medium at 30 °C for 24 h with shaking at 200 rpm. Then, the cells were spun down and transferred to 250 ml of SD medium without leucine and cultured for 24 h at 30 °C with shaking at 200 rpm. Next, 50 ml of preculture was transferred to 800 ml of expression medium and cultured at 30 °C for 24 h with shaking at 200 rpm. Then, 200 ml of induction medium was applied to the cell culture and the culture continued at 25 °C for 24 h to induce protein expression. Finally, the cells were harvested at 8,000$g$; the cell pellet was washed with lysis buffer and then frozen in a high-pressure homogenizer (Xpress) at −20 °C for 16 h.

SD medium: 20 g L$^{-1}$ glucose, 1.9 g L$^{-1}$ yeast nitrogen base, 5 g L$^{-1}$ $(NH_4)_2SO_4$, 60 mg L$^{-1}$ leucine and 30 mg L$^{-1}$ lysine.

Expression medium: 3% (v/v) glycerol, 5 g L$^{-1}$ glucose, 1.9 g L$^{-1}$ yeast nitrogen base, 5 g L$^{-1}$ $(NH_4)_2SO_4$, 1.17 g L$^{-1}$ amino acid mixture with isoleucine and uracil dropout.

Induction medium: 3% (v/v) glycerol, 20 g L$^{-1}$ galactose, 1.9 g L$^{-1}$ yeast nitrogen base, 5 g L$^{-1}$ $(NH_4)_2SO_4$, 1.17 g L$^{-1}$ amino acid mixture with isoleucine and uracil dropout.

### Protein purification

Batches of cells were cultured and purified individually to capture different conformations. For the closed CtMrs2 state, frozen cells were disrupted using a high-pressure X-Bomb cell disrupter and resuspended to 50 mg ml$^{-1}$ with lysis buffer supplemented with 5 mM MgCl$_2$. Crude membranes were collected through initial low speed centrifugation at 4,000$g$ to remove unbroken cells, followed by ultracentrifugation at 165,000$g$. The obtained membranes were solubilized at 50 mg ml$^{-1}$ in buffer A supplemented with 2 mM MgCl$_2$ and 1% (w/v) n-dodecyl-β-maltoside (DDM, Anatrace) at 18 °C for 2 h and using slow rotation of the sample. The solubilized material was collected through ultracentrifugation at 190,000$g$ for 30 min. The protein was purified using a 5-ml Histrap immobilized metal affinity chromatography (IMAC) column (Cytiva) equilibrated with buffer A supplemented with 0.05% (w/v) DDM and 2 mM MgCl$_2$. Following sample binding, the column was washed with 20 column volumes of buffer A supplemented with 0.05% (w/v) DDM, 2 mM MgCl$_2$ and 60 mM imidazole to remove contaminants. The target protein was eluted with buffer A supplemented with 0.05% (w/v) DDM, 2 mM MgCl$_2$ and 300 mM imidazole. Relevant elution fractions were concentrated and applied to a Superose 6 SEC column (Cytiva) for further purification using buffer A supplemented with 0.05% (w/v) DDM and 2 mM MgCl$_2$. The protein purity was assessed using SDS–PAGE. The peak fractions were pooled and concentrated to around 5 mg ml$^{-1}$ for downstream

processing. For the open Mrs2 state, a similar procedure was applied but using 2 mM EDTA instead of Mg$^{2+}$ to the buffer except during the IMAC purification (where no Mg$^{2+}$ was applied for the preparation of the open state). For the samples used for the limited proteolysis assay and ITC experiments, the final SEC buffer was similar to buffer A but the pH was adjusted to 8.0.

Lysis buffer: 10 mM Tris-HCl pH 7.5, 60 mM NaCl and 10% (v/v) glycerol.

Buffer A: 50 mM HEPES–NaOH pH 7.5, 150 mM NaCl and 10% (v/v) glycerol.

### Nanodisc reconstitution

MSP1D1 lacking the His-tag was purified as previously reported[57,58]. POPC lipids (Avanti Lipids) were prepared in 20 mM Tris-HCl pH 7.5, 100 mM NaCl and 0.5% (w/v) DDM for the nanodisc reconstitution. The purified protein was reconstituted into MSP1D1 nanodiscs with a molar ratio of 1:5:75 of CtMrs2, MSP1D1 and lipids, respectively, and 200 mg of SM2 biobeads (Bio-Rad) were used to remove detergent for protein incorporation into the nanodiscs. For the closed state of CtMrs2, the assembled nanodisc sample was purified on a Superdex 200 column (Cytiva) with buffer containing 50 mM HEPES–Na pH 7.5, 100 mM NaCl and 2 mM MgCl$_2$ and then TEV protease was added into the corresponding pooled peak fractions before incubating for 4 h at 4 °C. The cleaved CtMrs2 disc sample lacking GFP was further polished using a second SEC purification with same buffer. The protein-containing corresponding peak fractions were pooled and concentrated to 0.9 mg ml$^{-1}$ for cryo-EM grid preparation. For the open state of CtMrs2, the reconstituted sample was directly purified by SEC using a Superose 6 (Cytiva) column with running buffer containing 50 mM HEPES pH 7.5, 100 mM NaCl and 2 mM EDTA. The protein-containing peak fractions were pooled and concentrated to 8 mg ml$^{-1}$ for cryo-EM grid preparation.

### Cryo-EM grid preparation and data acquisition

For the closed state of CtMrs2, the CtMrs2 disc sample was incubated with an additional 10 mM MgCl$_2$ on ice for 16 h. Then, 3 µl of purified CtMrs2 disc sample (0.9 mg ml$^{-1}$) was applied to glow-discharged Quantifoil Cu R 1.2/1.3 300-mesh holey carbon grids, incubated for 3 s and blotted for 3 s at 4 °C and 100% humidity. Next, the grids were plunge-frozen into liquid ethane using a Vitrobot Mark IV. For the open state of CtMrs2, 1 mM fluorinated Fos-choline-8 was added to the purified CtMrs2 disc sample (8 mg ml$^{-1}$) sample immediately before preparation of the grids. Then, 3 µl of sample was applied to glow-discharged C-flat Cu R 1.2/1.3 300-mesh holey carbon grids, with a wait time of 3 s, blot force of 0 and blot time of 3 s. The grids were plunge-frozen into liquid ethane using a FEI Vitrobot Mark III at 4 °C and 100% humidity. The cryo-EM datasets were collected at the SciLifeLab Cryo-EM facility on a FEI Titan Krios EM instrument operated at an acceleration voltage of 300 kV equipped with a K3 (Gatan) detector with an applied energy filter of 20 eV and a magnification of 105,000. The dataset of the closed state was collected while operating in super-resolution mode bin 2 with a pixel size of 0.8617 Å and a total dose of 50 e$^-$ per Å$^2$ in 40 frames with a defocus range of −0.6 µm to -2.2 µm. The dataset of the open state was collected while operating in super-resolution mode bin 2 with a pixel size of 0.8566 Å and a total dose of 50 e$^-$ per Å$^2$ in 40 frames with a defocus range of −1.0 µm to -2.2 µm.

### Cryo-EM data processing

Data processing was performed using cryoSPARC[59], and the movies were initially processed using path motion correction (M), followed by Patch contrast transfer function (CTF) estimation. Bad micrographs, such as those that were empty, broken or containing ice, were removed through manual inspection.

For the closed state, 583 of 832 selected micrographs were first used for the blob picking and 294,000 particles were initially picked. The particles were extracted with bin 2 and subjected to two rounds

of 2D classification, resulting in 237,737 good particles. Then, 30,000 particles were used for the ab initio reconstruction, followed by heterogenous refinement with all particles, generating five classes. A second-round heterogenous refinement was performed with selected particles from classes 0 to 2, applying all five class maps, generating a map from 79,066 particles.

The selected templates from the first processing step were used for template picking based on 1,386 of 2,018 selected micrographs. A total of 1,316,900 particles were initially picked and extracted with bin 2 and then subjected to four rounds of 2D classification, resulting in 425,686 good particles. Then, 40,000 particles were used for the ab initio reconstruction, followed by heterogenous refinement with all particles, generating five classes. Four further rounds of heterogenous refinement were performed with selected particles from classes 0, 1 and 3, applying all five maps, generating a map based on 143,512 particles. Then, good particles from part 1 and part 2 were reextracted by applying local motion (M) correction. The final set of 222,578 particles and the resulting model were subjected to nonuniform refinement without imposing symmetry[60]. Then, CTF refinement was applied, followed by a final nonuniform refinement applying $C1$ or $C5$ symmetry, yielding two maps with an overall resolution of 3.1 Å ($C1$) and 2.7 Å ($C5$), based on a Fourier shell correlation (FSC) cutoff of 0.143. The local map resolution of 2.7 Å was estimated using cryoSPARC with an FSC cutoff of 0.143. The processing flow chart is shown in Extended Data Fig. 3.

For the open state, 200,519 particles were initially picked from 835 of 6,484 micrographs through blob picking. The particles were extracted and subjected to 2D classification to generate templates. The selected templates were used for template picking and a total of 3,058,006 particles were picked from 6,484 micrographs. Extracted particles were subjected to four rounds of 2D classification, resulting in 332,383 particles. A total of 40,000 particles were used for the ab initio reconstruction, followed by heterogenous refinement with all particles, generating five classes. A second round of heterogenous refinement was performed with selected particles from class 0 and class 4, respectively. The final set of 116,609 particles from class 0 and the resulting map were subjected to nonuniform refinement and CTF refinement without imposing symmetry. The final nonuniform refinements were performed without or with $C5$ symmetry, yielding two maps with overall resolutions of 3.5 Å ($C1$) and 3.2 Å ($C5$), based on an FSC cutoff of 0.143. The local map resolution of 3.2 Å was estimated using cryoSPARC with an FSC cutoff of 0.143. The processing flow chart is shown in Extended Data Fig. 8.

### Model building and refinement

The CtMrs2 AlphaFold[61] single-chain model (AF-G0S186-F1) (Supplementary Fig. 1a) was used as a template and initially fitted into the 2.7-Å cryo-EM density map of the closed state of CtMrs2 using USCF chimera[62]. Parts of the termini without visible cryo-EM density were removed in Coot[63] and de novo model building of the monomer, guided by the density for bulky side chains, was conducted in Coot. The monomer model was refined using real_space_refine[64] in PHENIX. The refined model was used to fit into the remaining cryo-EM density map of the remaining monomers. Several cycles of model building and adjustments in Coot and real-space refinement using real_space_refine[64] in PHENIX using the sharpened 2.7-Å map were performed to obtain the final model of the closed pentamer. For the open state, the same strategy was applied by starting with the refined model of the closed state of CtMrs2 as the initial template to fit into the 3.2-Å cryo-EM density map. Next, several cycles of model building and manual adjustments were performed in Coot and refinements were conducted using real_space_refine in PHENIX using the sharpened 3.2-Å map to obtain the final model. The final model was well resolved from A162 to V482, except for residues 262–270 connecting α3 and α4, thus displaying essentially the same sequence coverage as the closed structure. Model

validation was performed using MolProbity[65]. The figures were prepared with UCSF ChimeraX[66], UCSF Chimera and PyMOL.

### Yeast strains, manipulations and the complementation assay

Gene deletion was carried out through a PCR-based gene knockout strategy using hphNT1 selection antibiotic markers to generate an *MRS2*-deleted yeast strain. Mrs2-specific products from PCR amplification of hphNT1 were transformed into BY4741 WT yeast cells using herring sperm DNA in lithium acetate containing PEG3350 buffer. Antibiotic selection was achieved using 250 µg L$^{-1}$ hygromycin B (Invitrogen). Crude DNA extracts were used for PCR confirmation of *MRS2* gene deletion.

The full-length and mutation-complementary DNA of the *MRS2* gene were cloned from the genome of the BY4741 strain and constructed into multiple-copy p426 vectors. All mutations and full-length constructs were generated using a PCR-based strategy. The constructed Mrs2 plasmids were transferred to *MRS2*-deleted yeast cells using herring sperm DNA in lithium acetate containing PEG3350 buffer.

Yeast strains included WT BY4741 cells, *MRS2*-deleted BY4741 cells and *MRS2*-deleted BY4741 cells expressing different versions of GFP-tagged ScMrs2. Yeast cells were cultured using standard YPD medium (2% v/v glucose, 1% w/w yeast extract and 2% w/w bacteriological peptone) for 16 h at 30 °C and then inoculated in fresh YPD medium and cultured to the log phase. Half of the cells were collected and lysed with HU lysis buffer at 65 °C for 20 min to extract whole-cell content and measured for plasmid expression by immunoblotting. The remaining cells at an optical density (OD) of 1 was spotted on YPD plates (2% v/v glucose, 1% w/v yeast extract, 2% w/v bacteriological peptone and 2% w/v agar) and YPG plates (2% v/v glycerol, 1% w/v yeast extract, 2% w/v bacteriological peptone and 2% w/v agar) with a gradient dilution. Following 4 days of growth at 30 °C, the plates were scanned. Mouse monoclonal antibodies to the HA epitope (sc-7392) were purchased from Santa Cruz Biotechnology and used at 1:2,000 dilution for the experiment. Mouse monoclonal antibody to PKG1 (ab113687) was purchased from Abcam and used at 1:10,000 dilution for the experiment.

### Ni$^{2+}$ sensitivity assay

For the Ni$^{2+}$ sensitivity assay, *E. coli* BL21(DE3) cells were transformed with empty pET-22b plasmid or plasmid with CtMrs2 forms. The constructs were then transformed spread on LB agar plates supplemented with a working concentration of 50 µg ml$^{-1}$ ampicillin (throughout) and grown at 37 °C for 16 h. Single colonies were inoculated at 37 °C in LB medium supplemented with ampicillin for approximately 4 h until the OD of the cells reached 0.65. The cell cultures were then serially tenfold diluted with LB medium supplemented with ampicillin. Next, 6-µl drops of cells were spotted onto LB agar plates supplemented with ampicillin and IPTG (final concentration 10 µM) and the indicated amounts of Ni$^{2+}$ (0 or 1.4 mM) or Mg$^{2+}$ (0 or 1 mM). The plates were incubated at 37 °C for 20 h to compare the colony growth. Plates were imaged using the ChemiDoc MP Imaging System (Bio-rad).

### Limited proteolysis assay

Purified WT CtMrs2 fused with GFP and mutant forms thereof were used for the assay (2 mg ml$^{-1}$ stock solutions). Next, 10 µg of protein was incubated on ice or at 18 °C for 30 min with the desired amount of Mg$^{2+}$ or EDTA with a final reaction volume of 12 µl. Then, 1 µl of trypsin or chymotrypsin (0.1 mg ml$^{-1}$) was added to each sample, yielding an approximate molar ratio of 1:100 of protein and protease, respectively, and the reactions were incubated at 4 °C for 14 h or at 37 °C for 1 h. Then, 4 µl of loading buffer (4×) was applied to the reaction samples before incubating at 98 °C for 6 min. The samples were assessed using 12% SDS–PAGE Bis–Tris precasted gels and stained with Coomassie blue. The gels were imaged using the ChemiDoc MP Imaging System (Bio-rad).

## Xenopus laevis oocyte experiments

The WT construct employed for the cryo-EM study and corresponding mutants were subcloned into the pGEM expression plasmid and all the constructs were confirmed by sequencing (Macrogen). Complementary RNA (cRNA) was prepared using the T7 mMessage mMachine transcription kit (Ambion, Invitrogen). The RNA concentration was quantified using spectrophotometry (NanoDrop 2000c; Thermo Fisher Scientific). *Xenopus* oocytes were surgically isolated at Linköping University. The surgical procedure of *X. laevis* frogs was approved by the Linköping Animal Care and Use Committee (permit no. 14515) and conforms to national and international guidelines. Isolated *Xenopus* oocytes were injected with 50 nl of cRNA with a concentration of $1 \mu g \mu l^{-1}$ and incubated at 8 °C for 3–5 days followed by 16 °C for 1 day in modified Barth's solution consisting of 88 mM NaCl, 1 mM KCl, 2.4 mM NaHCO$_3$, 0.33 mM Ca(NO$_3$)$_2$, 0.41 mM CaCl$_2$, 0.82 mM MgSO$_4$, 15 mM HEPES and 2.5 mM pyruvate, with pH set to 7.6 using NaOH. To allow for time-matched comparisons of current amplitude, WT and mutant Mrs2 were injected and incubated under identical conditions.

Two-electrode voltage clamp recordings were performed at 18 °C using a Dagan CA-1B amplifier. Pulled microelectrodes (0.4–1.5 MΩ; World Precision Instruments) were filled with 3 M KCl. Whole-cell currents were sampled using Clampex (Molecular Devices) at 5 kHz and filtered at 500 Hz. For illustrative reasons, in figures, the currents were filtered further to minimize noise from the perfusion system. The holding voltage was set to −60 mV and all recordings were performed with the membrane potential clamped at −60 mV. Oocytes were initially perfused extracellularly with a Mg$^{2+}$-free control solution containing 100 mM *N*-methyl-D-glucamine and 10 mM HEPES, with pH set to 7.4 using HCl. To test for Mg$^{2+}$-induced currents, the extracellular perfusion solution changed to a Mg$^{2+}$-supplemented solution containing 80 mM *N*-methyl-D-glucamine, 20 mM MgCl$_2$ and 10 mM HEPES, with pH set to 7.4 using HCl. The content of the Mg$^{2+}$-free and Mg$^{2+}$-supplemented solutions was guided by a previous study[41]. The Mg$^{2+}$-free or Mg$^{2+}$-supplemented solution was continuously perfused through the recording chamber (0.5 ml min$^{-1}$) using a pump (Harvard Apparatus MP II, CMA Microdialysis).

Electrophysiological analysis was performed in GraphPad Prism 10 (GraphPad Software). The peak amplitude of the inward current was quantified by subtracting the stable basal current during initial perfusion with Mg$^{2+}$-free solution (likely caused by a minor unspecific leak upon penetrating the oocyte with the two electrodes) from the peak current during perfusion with Mg$^{2+}$-supplemented solution. The extent of current decay during perfusion with Mg$^{2+}$-supplemented solution was quantified by calculating the difference in current amplitude between the peak current and the current amplitude at the end of the 2-min period that Mg$^{2+}$ was supplied (analysis details in Supplementary Fig. 2). The ability of 1 mM cobalt hexamine to reduce the inward current was determined by calculating the difference in current amplitude without and with 1 mM cobalt hexamine added to the Mg$^{2+}$-supplemented solution. Only oocytes with a basal and stable leak current < 200 nA were included for analysis.

Statistical analysis was performed in GraphPad Prism 10 (GraphPad Software). Data are shown as the mean ± s.e.m. Statistics involving several groups were calculated using a one-way analysis of variance (ANOVA) followed by a Dunnett's multiple-comparisons test to compare the data for mutants and the WT. Statistics for the pharmacological effects of cobalt hexamine were calculated using a paired *t*-test. A *P* value < 0.05 was considered significant. In Fig. 5d, $I_{peak}$ denotes the peak current amplitude subsequent to the perfusion of Mg$^{2+}$-containing solution (as described above). $\Delta I_{amp}$ (%) denotes the spontaneous current decay during the perfusion of Mg$^{2+}$-containing solution, which was quantified as the difference in current amplitude between the peak current and the remaining current at the end of the 2-min period of Mg$^{2+}$ perfusion (as described above). For $I_{peak}$, $n = 19$ (WT), 5 (S328A;T329A), 5 (S396A;397A) and 7 (E374A;E378A). For $\Delta I_{amp}$ (%), $n = 12$ (WT), 5 (S328A;T329A), 5 (S396A;397A) and 12 (E374A;E378A). Note that GraphPad plots overlapping individual values side-by-side, for clarity.

## ITC

ITC experiments were performed on a MicroCal PEAQ ITC instrument (Malvern Panalytical) at a temperature of 318.1 ± 0.1 K by titrating Mg$^{2+}$ at a concentration of 50 ± 1 mM (25 ± 1 mM) into the cell (cell volume: 200 μl), containing the protein at a monomer concentration of 25 ± 2 μM (determined by ultraviolet absorption at 280 nm using an extinction coefficient of 65,780 M$^{-1}$ cm$^{-1}$). The principles of the ITC method are described elsewhere[67]. The instrument conditions were a rotation of 750 rpm and differential power of 5 μcal s$^{-1}$. The GFP-fused CtMrs2 proteins in the form of WT and the four mutants (E374R, S328A;T329A, S396A;S397A and E374A;E378A) were used in the experiments. The protein and Mg$^{2+}$ solutions (MgCl$_2$) were prepared to achieve exact buffer-matching conditions with the two solutions having identical concentrations of all solutes except for the protein, Cl$^-$ and Mg$^{2+}$. Control experiments were performed before each experiment session by injecting 50 mM (25 mM) MgCl$_2$, 50 mM HEPES pH 8.0, 150 mM NaCl, 10% (v/v) glycerol and 0.03% (w/v) DDM into the buffer solution of 50 mM HEPES pH 8.0, 150 mM NaCl, 10% (v/v) glycerol and 0.03% (w/v) DDM. Initially, a Mg$^{2+}$ concentration of 1,000 μM was titrated into 15 μM WT CtMrs2 to test for high-affinity binding. The heat produced under these conditions was too small to be reliably quantified, with a $\delta(\Delta H)$ between the first and last injection of only -0.5 kJ mol$^{-1}$. The Mg$^{2+}$ concentration was consequentially increased fiftyfold to 50 mM to match the conditions of the protease digest assay and the electrophysiology measurements. For all ITC experiments, 18 or 19 injections of 2 μl of Mg$^{2+}$ were used, with duplicates for each complex except for the WT where one experiment at 50 mM and one at 25 mM Mg$^{2+}$ were run. The time between injections was set to 4 min to optimize the peak integration and the error analysis was performed in NITPIC to achieve individual error bars assigned to each injection[54,68]. Before the peak integration, the control was subtracted.

## Determination Mg$^{2+}$-binding stoichiometry using ICP-MS

The protein samples were purified in DDM detergent as outlined above. For the Mg$^{2+}$-bound WT sample, 2 mM Mg$^{2+}$ was present in all the buffers except the final size-exclusion purification step to remove excess Mg$^{2+}$. The two mutant samples (S328A;T329A and S396A;S397A) were prepared similarly to the WT sample. To generate the EDTA-treated sample, the protein was purified as mentioned above for the open state and further purified on a HiTrap desalting column to remove excess EDTA for ICP-MS analysis. Magnesium content ($^{24}$Mg, $^{25}$Mg and $^{26}$Mg isotopes) was quantified by ICP-MS (Agilent 7900, equipped with an autosampler) after digesting the CtMrs2 samples (CtMrs2 monomer concentration = 12–17 μM) in 67% HNO$_3$ (v/v) for 16 h at 85 °C and subsequently diluting each sample to a final concentration of 1% HNO$_3$ (v/v) using ultrapure H$_2$O. Final magnesium concentrations were calculated accounting for dilution (average of five technical replicates) before the determination of experimental Mg-to-protein ratios. Protein concentration was determined using a Bradford assay with BSA as the standard.

## Yeast strains used in this study

| | | |
|---|---|---|
| *MATa ura3Δ0 leu2Δ0 his3Δ1 met15Δ0* | EUROSCARF | BY4741(WT) |
| BY4741 *mrs2::hphNT1* | This paper | *mrs2Δ* |

Italic formatting represents gene replacement, deletion or mutation.

## Expression constructs used in this study

| | | |
|---|---|---|
| p426-prADH1-HA-Mrs2 | This paper | Mrs2$^{FL}$ |
| p426-prADH1-HA-Mrs2-E341K/E342K | This paper | Mrs2$^{E341K;E342K}$ |
| p426-prADH1-HA-Mrs2-GMN333-335AAA | This paper | Mrs2$^{G333A;M334A;N335A}$ |
| p426-prADH1-HA-Mrs2-S326A | This paper | Mrs2$^{S326A}$ |
| p426-prADH1-HA-Mrs2-T319A | This paper | Mrs2$^{T319A}$ |
| p426-prADH1-HA-Mrs2-M309A | This paper | Mrs2$^{M309A}$ |
| p426-prADH1-HA-Mrs2-RN305-306AA | This paper | Mrs2$^{R305A;N306A}$ |
| p426-prADH1-HA-Mrs2-D302A | This paper | Mrs2$^{D302A}$ |

### Reporting summary

Further information on research design is available in the Nature Portfolio Reporting Summary linked to this article.

### Data availability

Cryo-EM maps and atomic coordinates were deposited to the EM Data Bank and the PDB, respectively, under accession codes EMD-18256 and 8Q8P (the closed state) and EMD-18257 and 8Q8Q (the open state). The AlphaFold models used in this study were downloaded from the Alpha-Fold Protein Structure Database under accession codes AF-G0S186-F1 (CtMrs2) and AF-Q01926-F1 (ScMrs2). Source data are provided with this paper.

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

### Acknowledgements

This study was supported by the Royal Physiographic Society of Lund (F 2022/1942 to P.L.), Swedish Research Council (2016-04474 and 2022-01315 to P.G., 2021-01885 to S.I.L. and 202004888 to K.S.J.), Knut and Alice Wallenberg Foundation (2015.0131 and 2020.0194 to P.G. and 2022.0105 to P.G. and S.I.L.), Lundbeck Foundation (R313-2019-774 to P.G.), Danish Council for Independent Research (9039-00273 to P.G.), Carl Trygger Foundation (CTS 21:1773), Crafoord Foundation (20170818, 20180652, 20200739 and 20220905 to P.G. and 20230966 to K.S.J.), Per-Eric and Ulla Schyberg Foundation (38267 to P.G. and separately also to K.S.J.) and Åke Wiberg's Foundation (M23-0143 to K.S.J.). G.M. is supported by the National Institute of General Medical Sciences of the National Institutes of Health (R35GM128704), the Robert A. Welch Foundation (AT-2073-20210327 and AT-2073-20240404) and the National Science Foundation (CHE-2045984). The funders had no role in study design, data collection and analysis, decision to publish, or preparation of the manuscript. We would like to thank P. A. Pedersen for providing the PAP1500 *S. cerevisiae* strain and J. Conrad, K. Wallden and M. Corrani at the Cryo-EM Swedish National Facility in Stockholm for sample preparation, screening and data collection. The Cryo-EM Swedish National Facility at SciLifeLab is funded by the Knut and Alice Wallenberg, Family Erling Persson and Kempe Foundations, SciLifeLab, Stockholm University and Umeå University. We also acknowledge the Lund University Cryo-EM platform for access to the GPU computers.

### Author contributions

P.L. and P.G. supervised the project. P.L. initiated the project, designed the experiments, performed the cloning, established and optimized protein overproduction and purification and designed the nanodisc reconstitution for the cryo-EM studies. P.L. processed the cryo-EM data and built and refined the structures. P.L. also executed the *E. coli* Ni$^{2+}$ sensitivity and limited proteolysis assays. S.L. performed the yeast growth assay supervised by K.L. R.L.E.V. and G.M. conducted the ICP-MS measurements. P.H. performed the HOLE pathway analyses supervised by K.L.-P. S.I.L. performed the *Xenopus* oocyte electrophysiology experiments. J.W. conducted the ITC experiments supervised by K.S.J. and both J.W. and K.S.J. analyzed the ITC data. P.L. conducted the initial data analysis, wrote the first draft and prepared the figures. P.L. and P.G. further conducted data analysis and interpretation and finalized the manuscript. All authors commented on the manuscript.

### Funding

### Competing interests

The authors declare no competing interests.

### Additional information

**Extended data** is available for this paper at https://doi.org/10.1038/s41594-024-01432-1.

**Correspondence and requests for materials** should be addressed to Ping Li or Pontus Gourdon.

Peer reviewer reports are available. Primary Handling Editor: Katarzyna Ciazynska, in collaboration with the *Nature Structural & Molecular Biology* team.

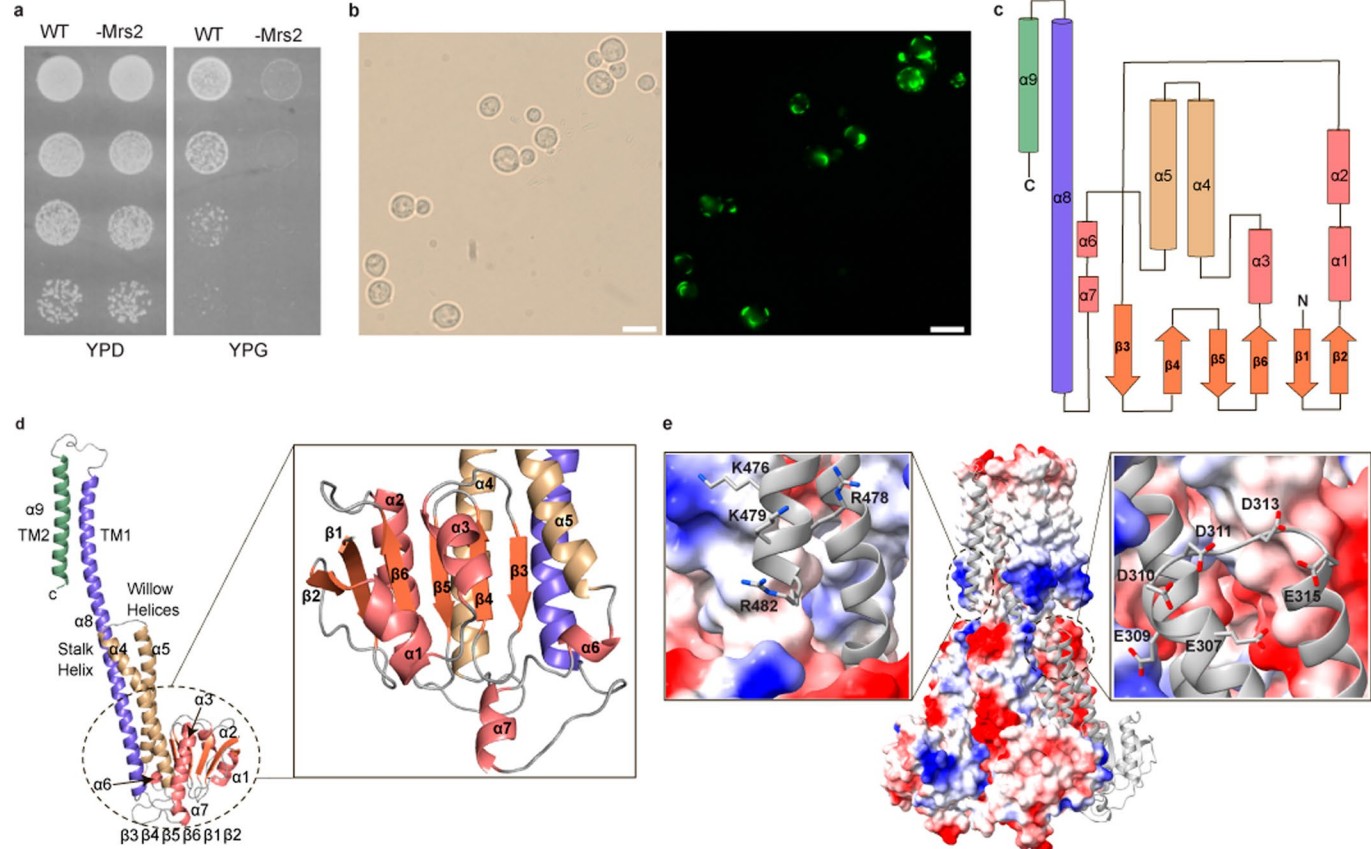

**Extended Data Fig. 1 | Functional characterization and secondary structure of CtMrs2. a**, Growth comparison of wild-type and Mrs2 knockout *S. cerevisiae* on fermentable carbon source (with glucose) and nonfermentable carbon source (glycerol). **b**, Live cell bioimaging micrographs of yeast cells expressing CtMrs2 with bright light (left) and GFP fluorescence (right), The scale bar represents 5 μm. The experiments were performed more than five times independently. The protein likely partitions to mitochondria. **c**, Schematic depiction of the secondary structure of a CtMrs2 monomer. **d**, Cartoon representation of the CtMrs2 monomer, colored as in panel c. **e**, Surface electrostatics of CtMrs2 with close-views of the basic and acidic rings.

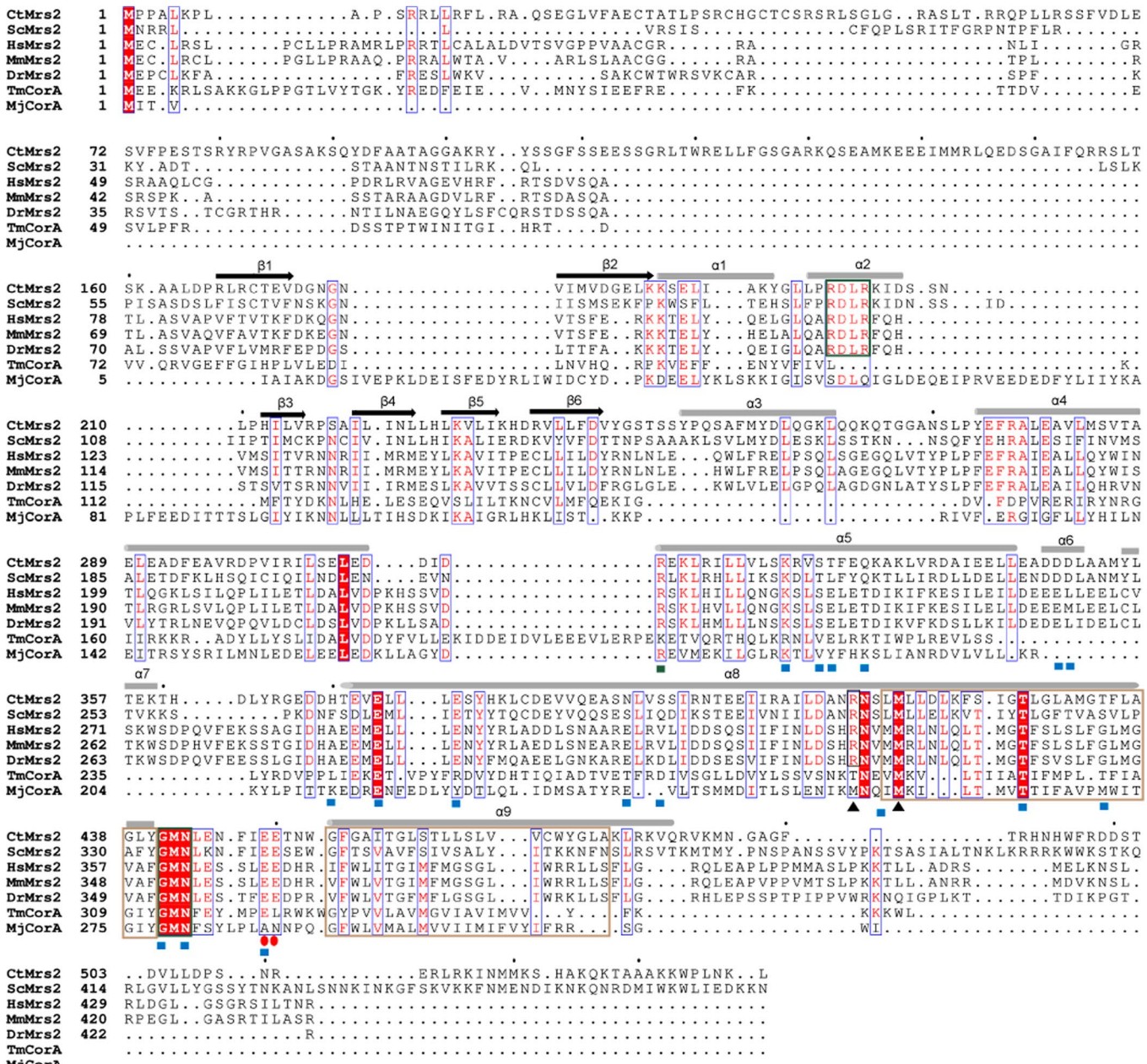

**Extended Data Fig. 2 | Sequence alignment of Mrs2 and CorA proteins.**
The following sequences are included in the alignment (Uniprot-ID in brackets): CtMrs2 from *Chaetomium thermophilum* (G0S186), ScMrs2 from *Saccharomyces cerevisiae* (Q01926), HsMrs2 (or hMrs2) from *Homo sapiens* (Q9HD23), MmMrs2 from Mus musculus (Mouse) (Q5NCE8), DrMrs2 from *Danio rerio* (E7F680), TmCorA from *Thermotoga maritima* (Q9WZ31) and MjCorA from *Methanocaldococcus jannaschii* (Q58439). There is relatively low sequence homology between Mrs2 and CorA proteins (for example 35 % between TmCorA and the equivalent from Saccharomyces cerevisiae, ScMrs2) and even among

Mrs2 members (for example 47 % between ScMrs2 and hMrs2). α-helices and β-strands are labelled with black arrows and grey cylinders as derived using the CtMrs2 structure. Black and blue arrowheads mark residues involved in gating (conserved arginine among Mrs2 proteins are highlighted with dark green box) and in Mg²⁺-binding, respectively. Green boxes denote the conserved GMN-motif selectivity filter and the conserved RDLR-regulation motif. Red circles represent residues of the conserved acidic loop. Wheat boxes show TM1 and TM2. Sequence alignments were performed using Cluster Omega (online) and visualized using ESPript 3.0 (online).

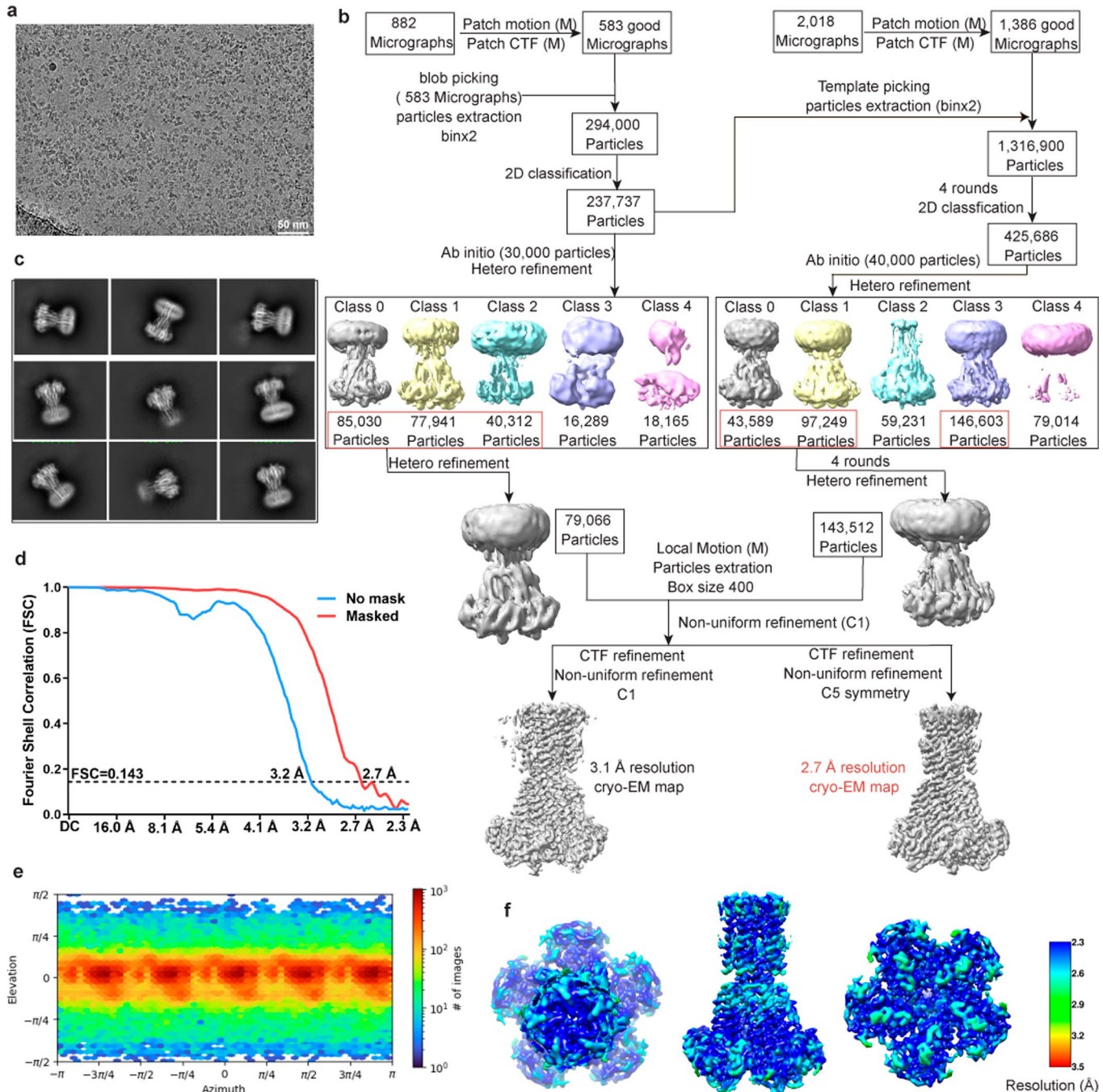

**Extended Data Fig. 3 | Cryo-EM data processing for the closed CtMrs2 state.**
**a**, Representative motion corrected micrographs, more than 2000 micrographs were collected. **b**, Cryo-EM data processing workflow. **c**, Selected 2D classes shows the orientation problem with most classes appearing as side-views and with only few tilted views. **d**, Gold standard Fourier shell correlation curves calculated for the reconstructed map based on a FSC 0.143 cut-off. **e**, Particle orientation distribution of the final reconstruction with predominant side-view particles. **f**, Local resolution maps with different views based on a FSC 0.143 cut-off.

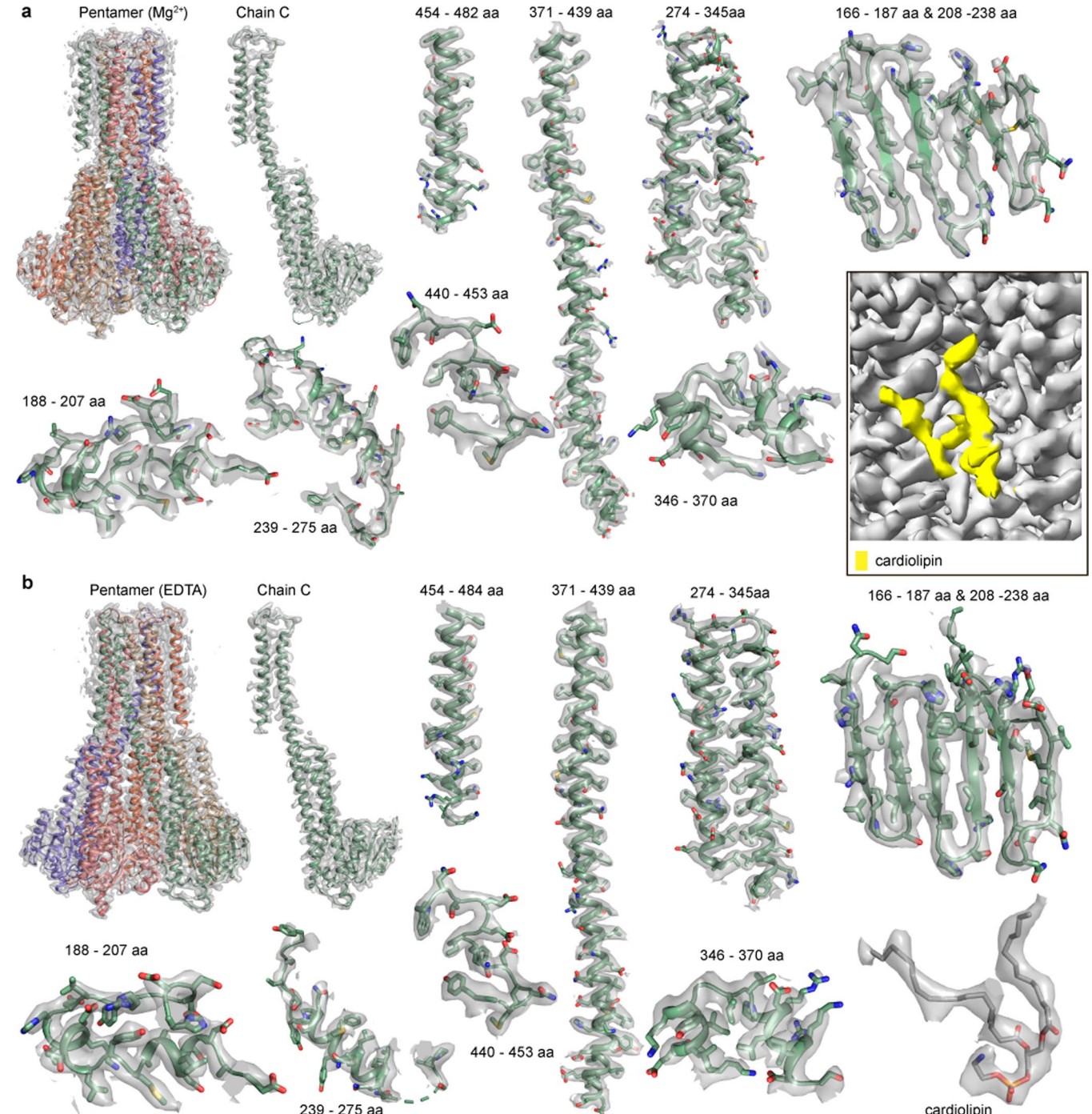

**Extended Data Fig. 4 | Cryo-EM density and associated model of representative parts of the closed and open CtMrs2 states. a**, The closed CtMrs2 pentamer, a separate monomer, and different segments. The cardiolipin density is colored in yellow. **b**, The open CtMrs2 pentamer, a separate monomer, and different segments. Interestingly, the open structure displays a clear lipid feature density in the groove between TM1 and TM2 of two adjacent monomers.

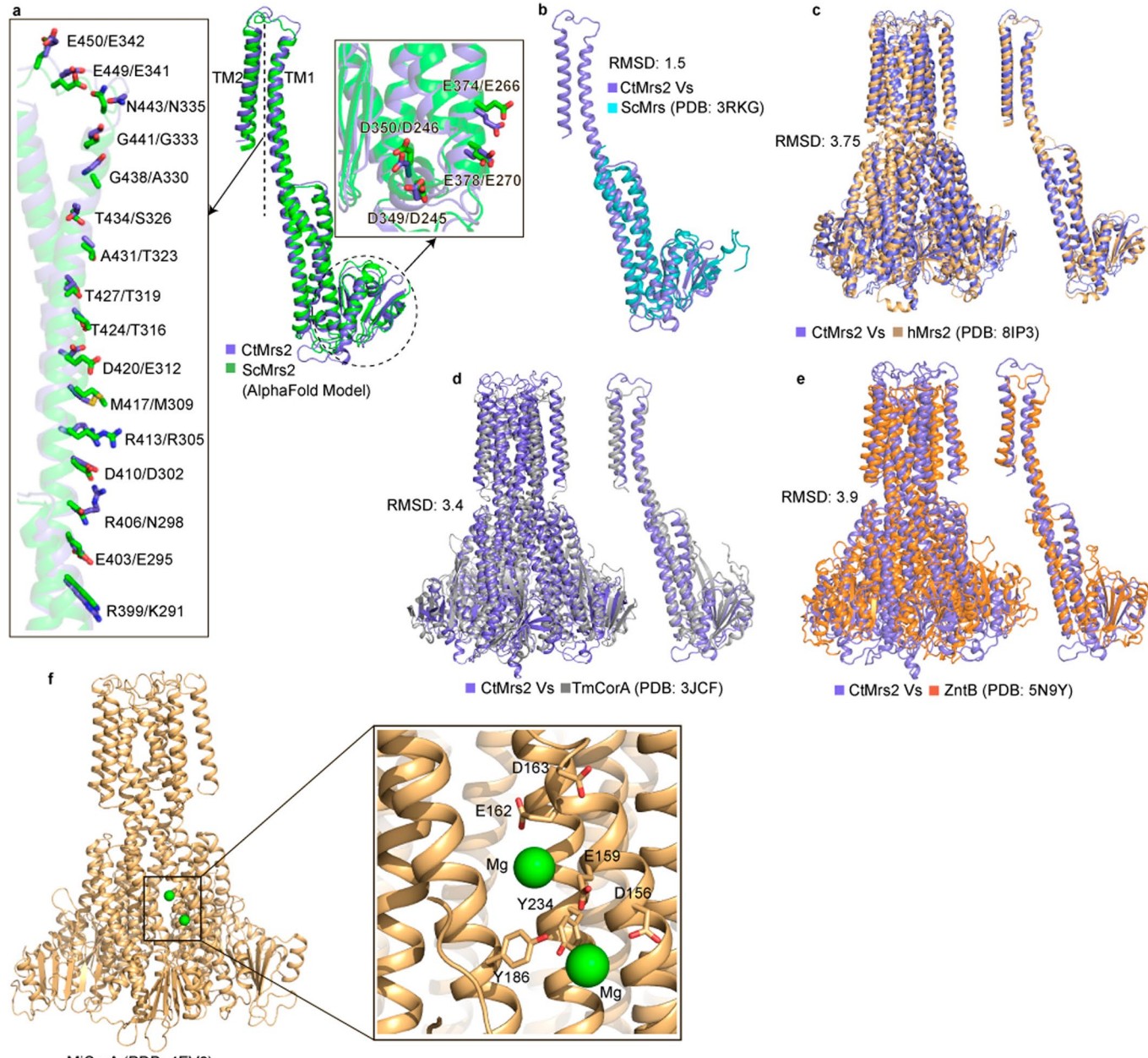

**Extended Data Fig. 5 | Structural comparisons of the closed CtMrs2 state with relevant conformations of ScMrs2, hMrs2, CorA, ZntB. a,** Alignment of a CtMrs2 monomer (blue) with the AlphaFold model of ScMrs2 monomer (green) (downloaded from AlphaFold Protein Structure Database), including a close-view of the residues lining the Mg$^{2+}$-conducting pathway and certain residues of the NTD. **b,** The soluble domain of CtMrs2 shows a similar fold as the one previously determined for the NTD of ScMrs2 (cyan, PDB-ID 3RKG). **c,** Alignments of CtMrs2 (blue) with human hMrs2 (wheat, PDB-ID 8IP3). The left panel shows an alignment of the pentamers, while the right panel displays an alignment of the monomers. **d,** Alignments of CtMrs2 (blue) with TmCorA (grey, PDB-ID 3JCF). The left panel shows an alignment of the pentamers, while the right panel displays an alignment of the monomers. **e,** Alignment of CtMrs2 (blue) with ZntB (orange, PDB-ID 5N9Y). The left panel shows an alignment of the pentamers, while the right panel displays an alignment of the monomers. **f,** Mg$^{2+}$-binding sites (corresponding to site M3 and M4 in CtMrs2) of MjCoA (wheat, PDB-ID 4EV6), and a close-view with residues shown as sticks, and with Mg$^{2+}$ shown as green spheres.

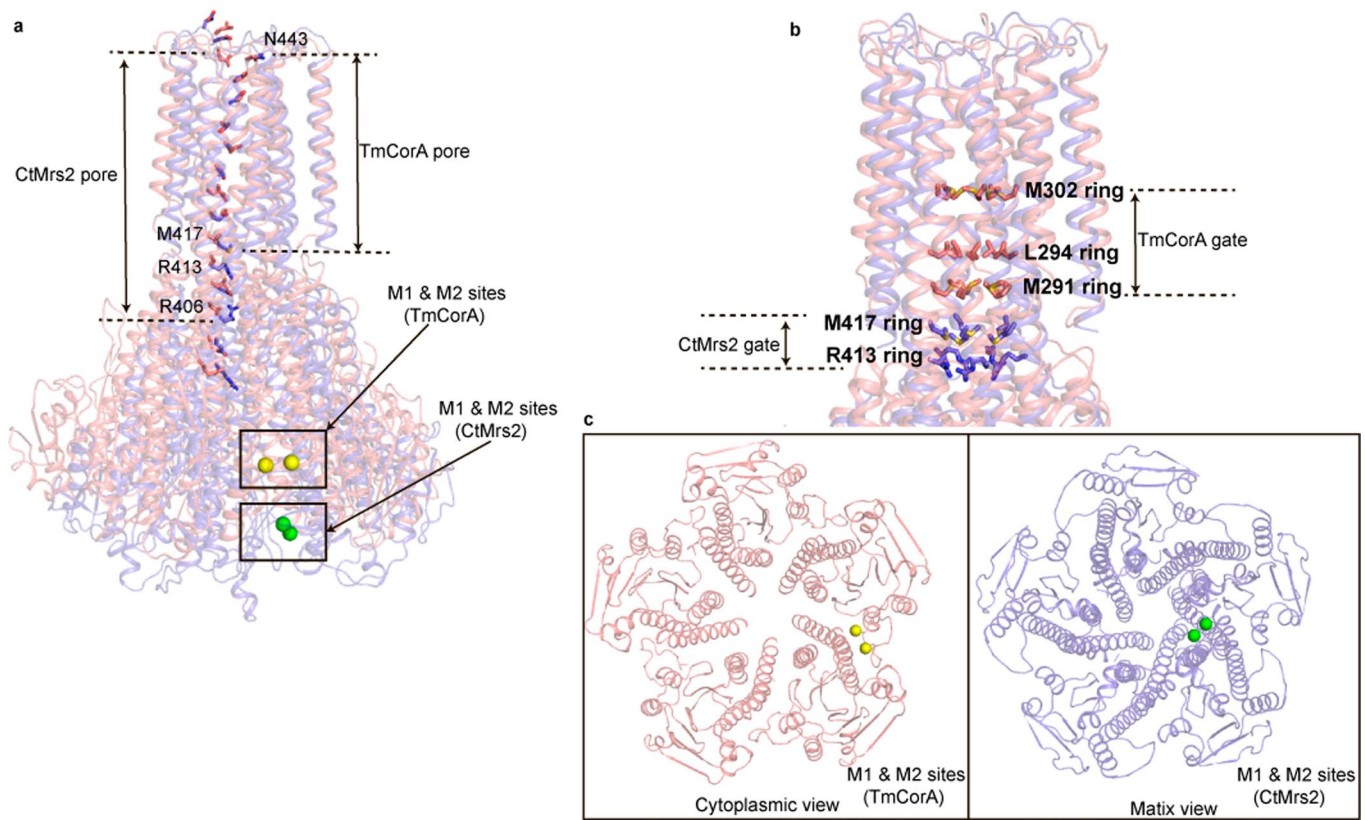

**Extended Data Fig. 6 | Structural comparisons of the ion conductance pathways and M1 & M2 sites of the closed states of CtMrs2 and TmCorA.** The figure was generated using alignment of the YGMN-motifs. CtMrs2 is shown in blue with Mg²⁺ at M1 and M2 as green spheres. TmCorA (PDB-ID 3JCF) is shown in pink with Mg²⁺ at M1 and M2 as yellow spheres. Residues lining the pore are shown as sticks and the approximate pore lengths are indicated. **a**, Overall view. **b**, Close-view of the gate residues for TmCorA and CtMrs2. **c**, Close-views of the M1/2 binding sites from the cytoplasm and mitochondrial matrix of TmCorA and CtMrs2, respectively.

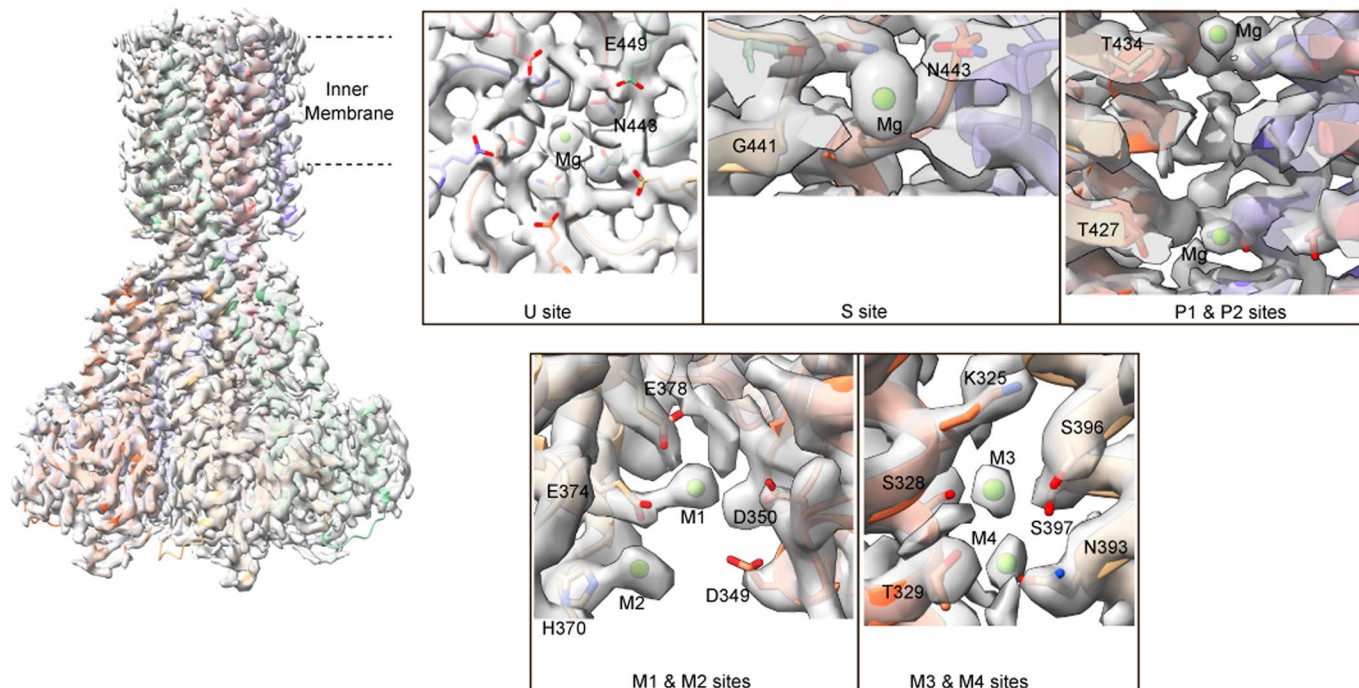

**Extended Data Fig. 7 | The cryo-EM C5 map of the closed state of CtMrs2 with details of the Mg²⁺-binding.** Side-view of CtMrs2 closed cartoon representation with associated cryo-EM map density (left). Close-views of the Mg²⁺-binding sites of the permeation pore (sites U, S as well as P1 and P2) as well as of the NTD in the mitochondrial matrix (sites M1 and M2 and M3 and M4).

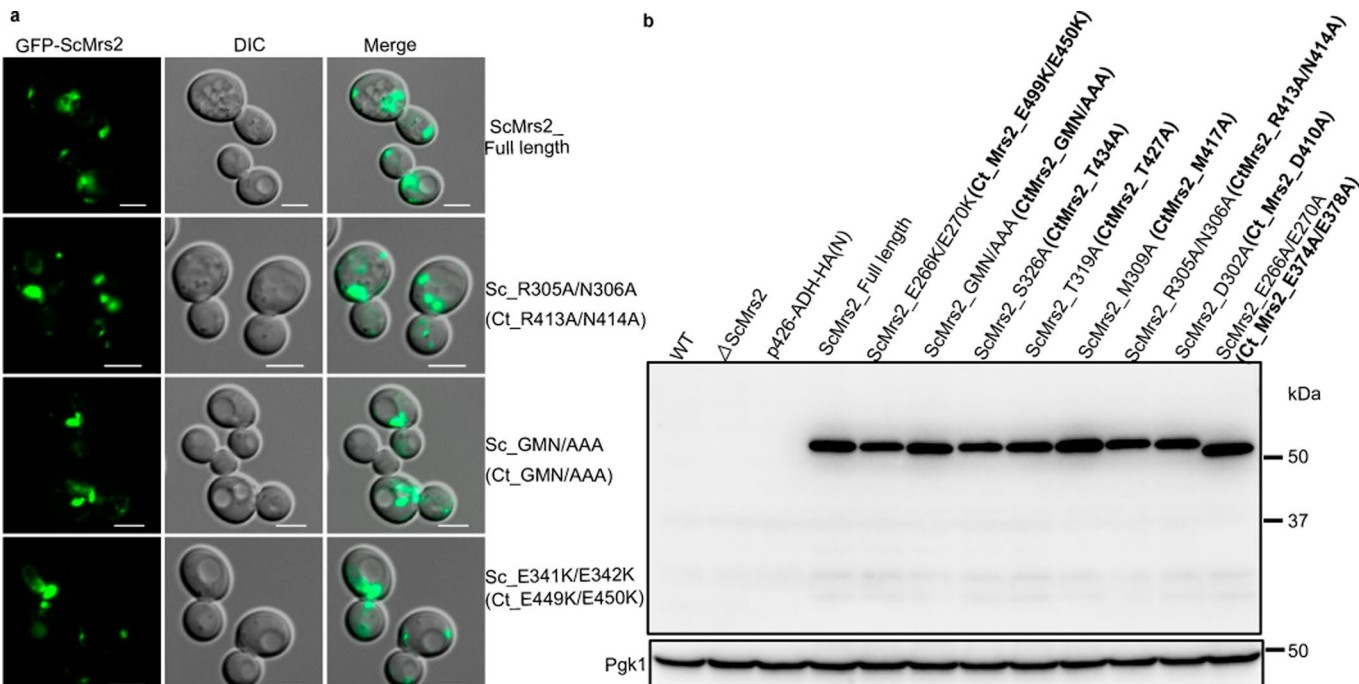

**Extended Data Fig. 8 | Validation of studied ScMrs2 forms in terms of cellular localization and expression.** WT represents the exploited *S. cerevisiae* strain (BY4741) used for the experiments. ΔScMrs2 is BY4741 lacking native Mrs2. P426-ADH-HA(N) is ΔScMrs2 transformed with empty vector. ScMrs2_Full length is ΔScMrs2 transformed with the P426-ADH-HA(N) vector harbouring full⁻length GFP-tagged ScMrs2, forming the basis for structure-function analyses. The corresponding variants of CtMrs2 are indicated in bold. **a**, The localization of ScMrs2 wild-type (ScMrs2_Full length) and mutants were investigated using fluorescence microscopy, comparing GFP-fluorescence (left), differential interference contrast (DIC) (middle) and merged (right), the scale bar represents 5 µm, Imaging experiments were conducted for three times with different clones of each yeast strain, and each time five different regions were captured containing around 100 cells in total. **b**, The expression of ScMrs2 wild-type (ScMrs2_Full length) and mutants probed using immunoblotting with an anti-HA antibody and using phosphoglycerate kinase (Pgk1) as a loading control. The experiments were conducted three times with different clones of each yeast strain.

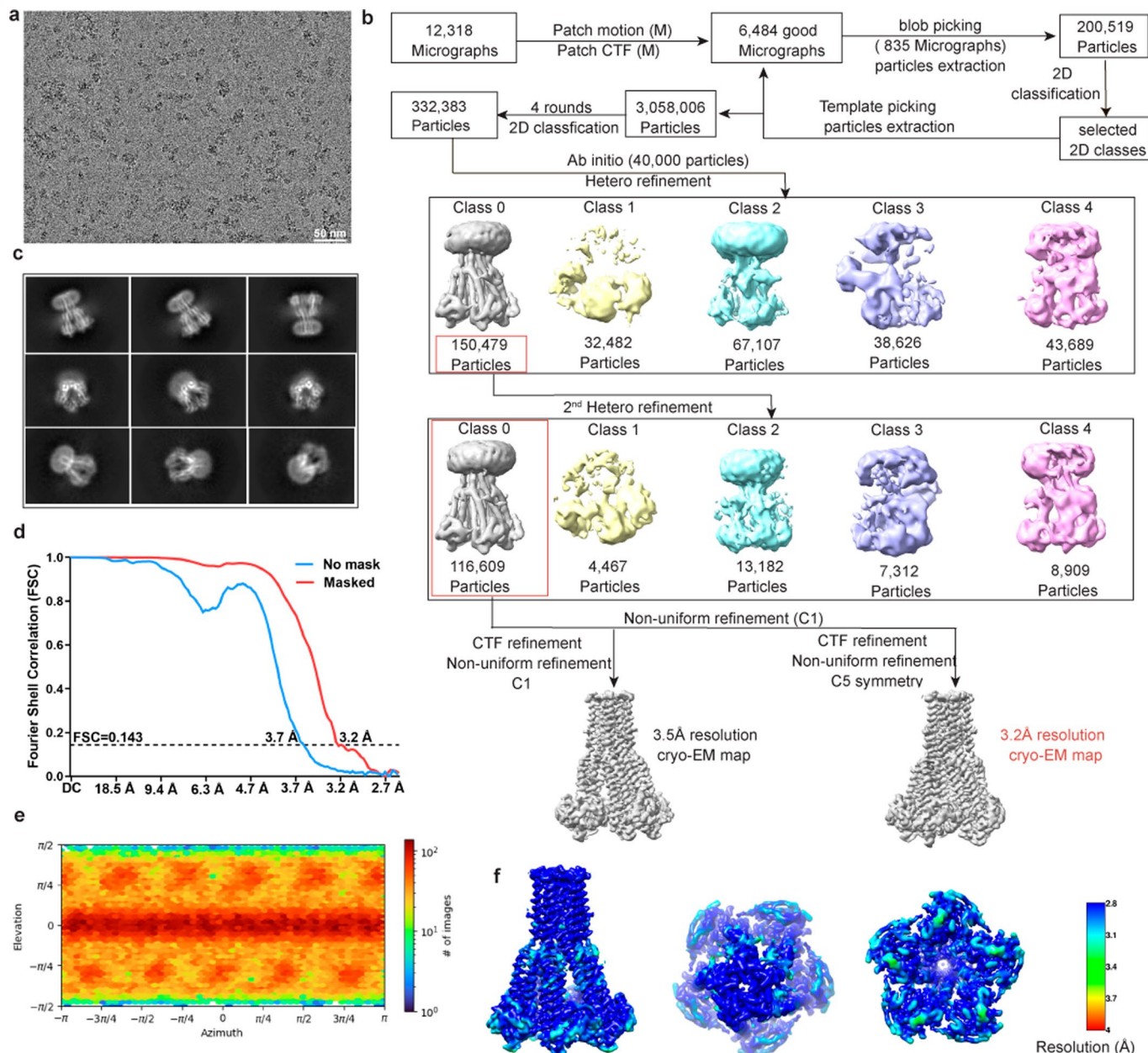

**Extended Data Fig. 9 | Cryo-EM data processing for the open CtMrs2 state.**
**a**, Representative motion corrected micrograph, more than 10000 micrographs were collected. **b**, Cryo-EM data processing workflow. **c**, Selected 2D classes with different views. **d**, Gold standard Fourier shell correlation curves calculated for the reconstructed map based on a FSC 0.143 cut-off. **e**, Particle orientation distribution of the final reconstruction. **f**, Local resolution maps with different views based on a FSC 0.143 cut-off.

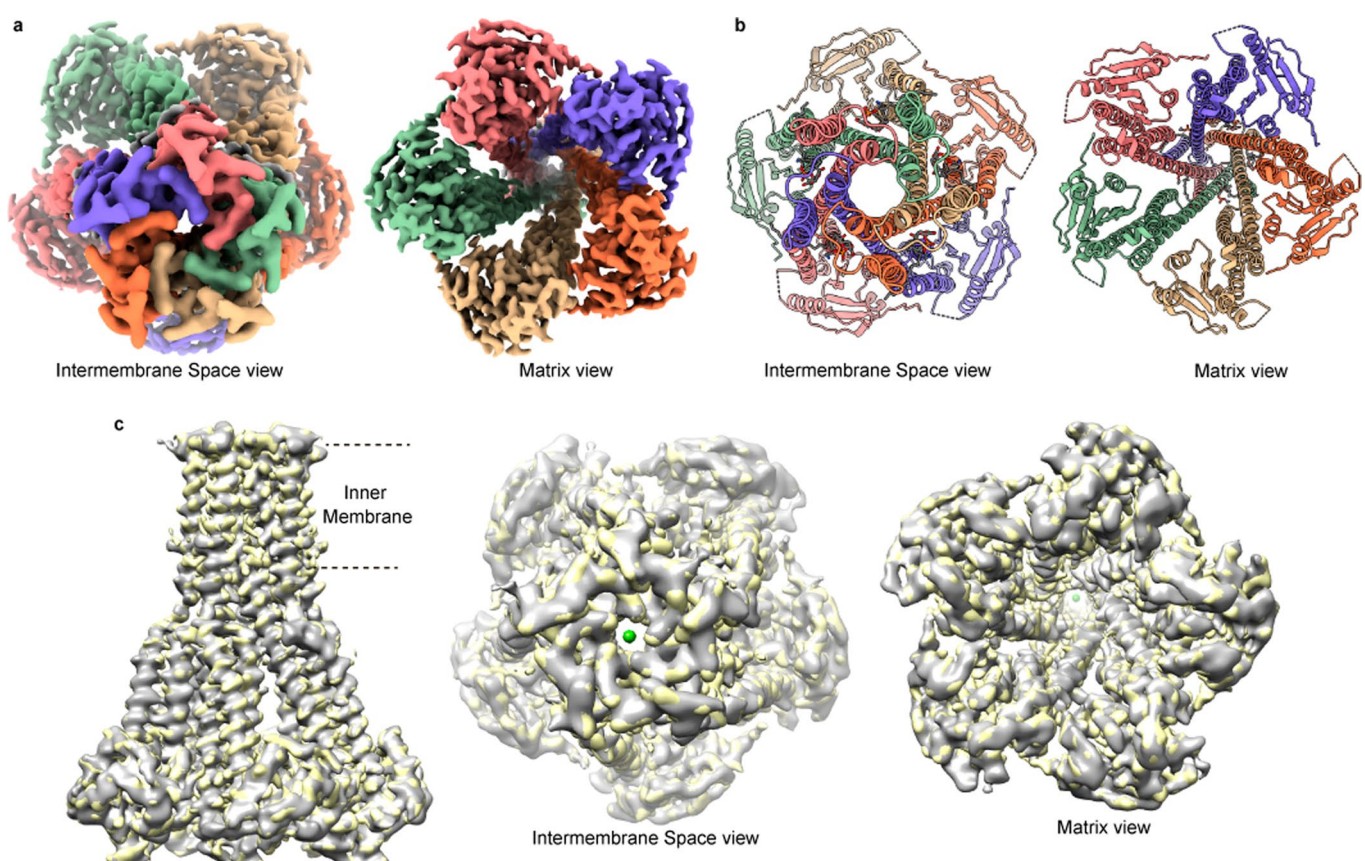

**Extended Data Fig. 10 | The open state of CtMrs2. a**, 3.2 Å overall resolution cryo-EM density of the CtMrs2 homo-pentamer (using C5 symmetry). The views are from the intermembrane space and from the matrix, respectively. The monomer are colored in blue, orange, wheat, green and pink. **b**, Cartoon representation of the CtMrs2 homo-pentamer with same views as in panel a and colored as in panel a. **c**, Comparison of two uncropped reconstructed cryo-EM maps, calculated in the absence of symmetry (C1, grey) or with C5 symmetry (yellow) imposed. The views are from the side, from the intermembrane space and from the matrix. The two maps overlay well. The $Mg^{2+}$ bound to GMN-motif selectivity filter is shown as a green sphere.

# Reporting Summary

## Statistics

For all statistical analyses, confirm that the following items are present in the figure legend, table legend, main text, or Methods section.

| n/a | Confirmed | |
|---|---|---|
| ☐ | ☒ | The exact sample size (*n*) for each experimental group/condition, given as a discrete number and unit of measurement |
| ☐ | ☒ | A statement on whether measurements were taken from distinct samples or whether the same sample was measured repeatedly |
| ☐ | ☒ | The statistical test(s) used AND whether they are one- or two-sided<br>*Only common tests should be described solely by name; describe more complex techniques in the Methods section.* |
| ☒ | ☐ | A description of all covariates tested |
| ☒ | ☐ | A description of any assumptions or corrections, such as tests of normality and adjustment for multiple comparisons |
| ☐ | ☒ | A full description of the statistical parameters including central tendency (e.g. means) or other basic estimates (e.g. regression coefficient) AND variation (e.g. standard deviation) or associated estimates of uncertainty (e.g. confidence intervals) |
| ☐ | ☒ | For null hypothesis testing, the test statistic (e.g. *F*, *t*, *r*) with confidence intervals, effect sizes, degrees of freedom and *P* value noted<br>*Give P values as exact values whenever suitable.* |
| ☒ | ☐ | For Bayesian analysis, information on the choice of priors and Markov chain Monte Carlo settings |
| ☒ | ☐ | For hierarchical and complex designs, identification of the appropriate level for tests and full reporting of outcomes |
| ☒ | ☐ | Estimates of effect sizes (e.g. Cohen's *d*, Pearson's *r*), indicating how they were calculated |

*Our web collection on statistics for biologists contains articles on many of the points above.*

## Software and code

Policy information about availability of computer code

| | |
|---|---|
| Data collection | Cryo-EM datasets were collected on Titan Krios electron microscopes with a Gatan K3 detector. $Mg^{2+}$ detection was performed by ICP-MS on Agilent 7900, equipped with an autosampler. ITC experiments were performed on a MicorCal PEAQ-ITC instrument (Malvern Panalytical) |
| Data analysis | Cryo-EM datasets were processed with cryosparc v3.3.2. Model building and refinement were performed using USCF chimera 1.14, Wincoot 0.9.2 and phenix 1.20.1. Figures were generated using USCF chimera v1.14, ChimeraX v1.5 and pymol v2.3.4. Sequence alignments were performed using Cluster Omega (online), and visualized using ESPript 3.0. ICP-MS data analysis was performed using GraphPad Prism 9 and oocyte experiments data analysis was performed using GraphPad Prism 10. ScMrs structure is predicted by AlphaFold2 and downloaded from Uniprot website. ITC experiments data analysis was performed using NITPIC software. |

For manuscripts utilizing custom algorithms or software that are central to the research but not yet described in published literature, software must be made available to editors and reviewers. We strongly encourage code deposition in a community repository (e.g. GitHub). See the Nature Portfolio guidelines for submitting code & software for further information.

# Data

Policy information about <u>availability of data</u>

All manuscripts must include a <u>data availability statement</u>. This statement should provide the following information, where applicable:
- Accession codes, unique identifiers, or web links for publicly available datasets
- A description of any restrictions on data availability
- For clinical datasets or third party data, please ensure that the statement adheres to our <u>policy</u>

> The sequence of Magnesium channel CtMrs2 is available in the following link:
> https://www.uniprot.org/uniprotkb/G0S186/entry
> Cryo-EM maps have been deposited in the Electron Microscopy Data Bank (EMDB) under accession codes:
> EMD-18256 (closed state) and EMD-18257 (open state).
> The atomic coordinates have been deposited in the Protein Data Bank (PDB) under accession codes 8Q8P (closed state) and 8Q8Q (open state).

# Research involving human participants, their data, or biological material

Policy information about studies with <u>human participants or human data</u>. See also policy information about <u>sex, gender (identity/presentation), and sexual orientation</u> and <u>race, ethnicity and racism</u>.

| | |
|---|---|
| Reporting on sex and gender | n/a |
| Reporting on race, ethnicity, or other socially relevant groupings | n/a |
| Population characteristics | n/a |
| Recruitment | n/a |
| Ethics oversight | n/a |

Note that full information on the approval of the study protocol must also be provided in the manuscript.

# Field-specific reporting

Please select the one below that is the best fit for your research. If you are not sure, read the appropriate sections before making your selection.

☒ Life sciences        ☐ Behavioural & social sciences        ☐ Ecological, evolutionary & environmental sciences

For a reference copy of the document with all sections, see nature.com/documents/nr-reporting-summary-flat.pdf

# Life sciences study design

All studies must disclose on these points even when the disclosure is negative.

| | |
|---|---|
| Sample size | The complete cryo-EM datasets were collected by microscopy with available time and each collected dataset was sufficient to obtain high resolution maps.<br>Five independent biological samples for different states were prepared for the ICP-MS study and protein concentration was determined by Bradford assay using BSA as a standard.<br>For electrophysiology experiments, each smaple group contained a minimum of 5 observations. During study design of oocyte experiments, the authors estimated what sample size to use based on the authors' extensive previous knowledge on how scattered these data usually are and how big differences are to be expected. |
| Data exclusions | In the cryo-EM data analysis, bad micrographs were excluded with low CTF fitting resolution and bad particles were excluded to generate the high resolution maps. |
| Replication | Multiple grids were prepared for each sample state and data was collected with selected grid for individual state. Samples for ICP-MS study were purified with 5 independent biological replicates. In vivo growth assay were repeated 3 times independently. or oocyte experiments, each experiment was replicated a minimum of 5 times. Replicate information is provided in appropriate figure legends. All replicates were successful, unless cells were excluded based on exclusion criteria described in the Methods (basal or unstable leak >200 nA)." |
| Randomization | No randomization applied in this study. For the cryo-EM data analysis, particles were processed automatically by Cryosparc. |
| Blinding | No blinding was applied in this study as no group allocation was used and blinding is not relevant to the cryo-EM analysis, ICP-MS data analysis and in vivo growth assay. |

# Reporting for specific materials, systems and methods

We require information from authors about some types of materials, experimental systems and methods used in many studies. Here, indicate whether each material, system or method listed is relevant to your study. If you are not sure if a list item applies to your research, read the appropriate section before selecting a response.

## Materials & experimental systems

| n/a | Involved in the study |
|---|---|
| ☐ | ☒ Antibodies |
| ☐ | ☒ Eukaryotic cell lines |
| ☒ | ☐ Palaeontology and archaeology |
| ☐ | ☒ Animals and other organisms |
| ☒ | ☐ Clinical data |
| ☒ | ☐ Dual use research of concern |
| ☒ | ☐ Plants |

## Methods

| n/a | Involved in the study |
|---|---|
| ☒ | ☐ ChIP-seq |
| ☒ | ☐ Flow cytometry |
| ☒ | ☐ MRI-based neuroimaging |

## Antibodies

| | |
|---|---|
| Antibodies used | Mouse monoclonal antibodies to against HA-epitope (sc-7392; 1:2000 for IB) were purchased from Santa Cruz Biotechnology, mouse monoclonal antibody to PGK1(ab113687; 1:10000 for IB) was purchased from Abcam. |
| Validation | Validated by negative controls without signal and positive controls show specific signals. |

## Eukaryotic cell lines

Policy information about cell lines and Sex and Gender in Research

| | |
|---|---|
| Cell line source(s) | S. cerevisiae (PAP1500) strain was used for the protein production. S. cerevisiae (BY4741) strain was used for growth assay. |
| Authentication | no authentication |
| Mycoplasma contamination | no |
| Commonly misidentified lines (See ICLAC register) | n/a |

## Animals and other research organisms

Policy information about studies involving animals; ARRIVE guidelines recommended for reporting animal research, and Sex and Gender in Research

| | |
|---|---|
| Laboratory animals | Adult Xenopus laevis frogs of an age of 24-55 months were used |
| Wild animals | No wild animals were used in the study. |
| Reporting on sex | Only female Xenopus laevis frogs were used, as only female animals produce oocytes. Note that experiments were performed only on isolated oocytes (no in vivo experiments were performed). |
| Field-collected samples | No field collected samples were used. |
| Ethics oversight | Oocyte isolation was approved by the Linköping Animal Care and Use Committee, as stated in the Methods section. |

Note that full information on the approval of the study protocol must also be provided in the manuscript.

