## [Peer review file · Nature Structural & Molecular Biology]

Closed and open structures of the eukaryotic magnesium channel Mrs2 reveal the auto-ligand-gating regulation mechanism

Corresponding Author: Dr Pontus Gourdon

This manuscript has been previously reviewed at another journal. This document only contains reviewer comments, rebuttal and decision letters for versions considered at Nature Structural & Molecular Biology.

Version 0:

Decision Letter:

10th Nov 2023

Dear Dr Gourdon,

Thank you again for submitting your manuscript "Closed and open structures of the eukaryotic magnesium channel Mrs2 reveal the auto-ligand-gating regulation mechanism". We now have comments (below) from the 2 reviewers who evaluated your paper. In light of those reports, we remain interested in your study and would like to see your response to the comments of the referees, in the form of a revised manuscript.

You will see that while reviewers appreciate the results, they raise several concerns which will need to be addressed in a revision. Specifically, we agree with the reviewers that further functional exploration of the role of M1-M4 sites is needed. Please also consider the use of ITC, as suggested by reviewer #2, to evaluate magnesium binding. Furthermore, in line with reviewer's #1 comments, please discuss and compare the published work from Shen and Matthies groups to put the results in better context (<https://doi.org/10.1038/s41467-023-40516-2> and <https://doi.org/10.1038/s41467-023-42599-3>).

Please be sure to address/respond to all concerns of the referees in full in a point-by-point response and highlight all changes in the revised manuscript text file.

We appreciate the requested revisions are extensive. We thus expect to see your revised manuscript within 6 months. If you cannot send it within this time, please let us know. We will be happy to consider your revision as long as nothing similar has been accepted for publication at NSMB or published elsewhere. Should your manuscript be substantially delayed without notifying us in advance and your article is eventually published, the received date would be that of the revised, not the original, version.

Reporting Summary:

Please note that all key data shown in the main figures as cropped gels or blots MUST be presented in uncropped form, with molecular weight markers. These data can be aggregated into a single supplementary figure. While these data can be displayed in a relatively informal style, they must refer back to the relevant figures.

SOURCE DATA: we urge authors to provide, in tabular form, the data underlying the graphical representations used in figures. This is to further increase transparency in data reporting, as detailed in this editorial (<http://www.nature.com/nsmb/journal/v22/n10/full/nsmb.3110.html>). Spreadsheets can be submitted in excel format. Only one (1) file per figure is permitted; thus, for multi-paneled figures, the source data for each panel should be clearly labeled in the Excel file; alternately the data can be provided as multiple, clearly labeled sheets in an Excel file. When submitting files, the title field should indicate which figure the source data pertains to.

We require deposition of coordinates (and, in the case of crystal structures, structure factors) into the Protein Data Bank with the designation of immediate release upon publication (HPUB). Electron microscopy-derived density maps and coordinate data must be deposited in EMDB and released upon publication. Deposition and immediate release of NMR chemical shift assignments are highly encouraged. Deposition of deep sequencing and microarray data is mandatory, and the datasets must be released prior to or upon publication. To avoid delays in publication, dataset accession numbers must be supplied with the final accepted manuscript and appropriate release dates must be indicated at the galley proof stage. Please find the complete NRG policies on data availability at <http://www.nature.com/authors/policies/availability.html>.

Link Redacted

Sincerely,

Katarzyna Ciazynska, PhD
(she/her)
Associate Editor
Nature Structural & Molecular Biology
<https://orcid.org/0000-0002-9899-2428>

Referee expertise:

Referee #1: Magnesium transport

Referee #2: cryo-EM, ion channels

Reviewers' Comments:

Reviewer #1:

Remarks to the Author:

In this work, the authors determined the cryo-EM structures of *Chaetomium thermophilum* Mrs2 in the presence and absence of Mg²⁺ and performed the structure-based mutational analysis. Importantly, the structure in the Mg²⁺-free condition seems to represent the open conformation, which has not been reported in previous structural studies of Mrs2. Furthermore, the structural comparison of two structures revealed the symmetrical conformational changes, which is different from those observed in the prokaryotic CorA protein. The manuscript is well written and the data presented are solid. Overall, I strongly recommend this manuscript for publication if the following concerns are adequately addressed.

Major concerns

1. Additional experiments to characterize the regulatory role of M1-M4 sites

The authors should verify the regulatory role of M1-M4 sites. The growth assay in Figure 2g is not appropriate to characterize this, as loss of the regulatory Mg²⁺ binding site would result in upregulation of Mg²⁺ uptake rather than loss of Mg²⁺ uptake. For example, the authors may try the Co²⁺ or Ni²⁺ sensitivity assay. Or the authors can also try limited proteolysis with purified CtMrs2 and its mutants of the M1-M4 sites in the presence and absence of Mg²⁺, as previously performed with TmCorA (PMID: 16902408). This is important because there is no previous functional report on CtMrs2. It is not clear even whether CtMrs2 is also an auto-ligand-gated ion channel like TmCorA. To properly interpret the Mg²⁺-dependent structural changes of CmMrs2 by cryo-EM, such functional experiments would be required.

2. Addition of discussion section

In the current manuscript, the section "Conductance and gating mechanisms in eukaryotic Mrs2 proteins" seems to correspond to the discussion section, while the discussion section is missing from the manuscript. The authors should expand this section as a discussion section. For example, it would be interesting to discuss the similarities and differences of the Mg²⁺-dependent regulation mechanism of Mrs2/CorA with that of other Mg²⁺ channels, which would further deepen the authors' findings.

3. Reason for the structural difference with the recent human Mrs2 structures

As the authors would be aware, very recently three independent groups reported the human Mrs2 structures in the presence and absence of Mg²⁺ ions (doi: <https://doi.org/10.1101/2023.08.22.553867>, doi: <https://doi.org/10.1101/2023.08.12.553106>, doi: <https://doi.org/10.1038/s41467-023-40516-2>). Interestingly, none of these structures showed the Mg²⁺-dependent structural changes at the conserved arginine (R413 in CtMrs2 and R332 in hMrs2) or the symmetric separation of the cytoplasmic domain. The authors should provide a reasonable explanation for this discrepancy. Is it because such structural changes are specific to CtMrs2 and not conserved in human Mrs2? If so, it would affect the significance of this work. Or does it reflect the different sample preparation conditions? All hMrs2 structures were determined in detergent, whereas CtMrs2 structures were determined in nanodisc. In addition, for hMrs2 structures in the presence of EDTA, EDTA was not added at the beginning of the purification for sample preparation.

Minor concerns

1. Supplementary Fig. 2.

Please include CorA proteins in the sequence alignment. It is important to show that Arg413 is not conserved in CorA.

2. Page 9. "Similarly, the G441A/M442A/N443A form exhibited reduced cell proliferation, which is consistent with a previous study demonstrating interruptions of the GMN-motif impair the protein function."

Please cite the relevant papers.

3. Fig. 2g

There is a typo. N14A should be corrected to N414A.

4. Legend of Fig. 2d. "Site U is positioned next to the loop in-between TM1 and TM2 and site S at the GMN-motif if the selective filter".

The part "if the selectivity filter" might be a typo? It should be corrected to "of the selectivity filter"?

5. Legend of Fig. 2d. "Sites T1 and T2 are located in the transmembrane domain. e, Close-views of sites M1-M4 in the N-terminal domain of CtMrs2".

T1 and T2 would be a typo and should be corrected as P1 and P2.

6. Fig. 3.

The authors should also illustrate the structural difference of R406, as it is the most constricted site in the closed state.

7. Fig. 4c, d.

The way the authors show M1/M2 (cartoon) and M3/4 (surface) is not consistent. Please show M1/M2 and M3/M4 in both cartoon and surface representations.

8. Supplementary Fig. 5a.

How did the authors obtain the AlphaFold model of ScMrs2? By making the model themselves? Or by downloading it from a database? Please clarify, and if they generated the model themselves, please provide the model and log files of AlphaFold.

Reviewer #2:

Remarks to the Author:

CorA/Mrs2 proteins play essential roles in cation permeation across cellular membranes, especially Mg²⁺ import into mitochondria matrix. How these channels work remained unclear in a large part due to the absence of conductive/open conformation structure. The authors present here near-atomic resolution structures of a eukaryote (fungus) Mrs2 protein in conformations consistent with both closed and open states in lipid nano-discs. The Mg²⁺ autoregulation site has been proposed. Overall, this work provides important new open structural information that promotes the understanding this family of proteins. Evaluation of functional relevance in the proposed Mg²⁺ regulation sites would further test the proposed working mechanism. With a few other minor clarifications/edits, this work should be shared with the community.

Major point:

1. Mg²⁺ binding and autoregulation. The same Mg²⁺:Mrs2 stoichiometry of ~2 across EDTA-treated/M3-M4 mutants/Mg²⁺ samples is interesting and also raises some concerns. Mg²⁺ was built in the S, P1 and P2 sites even in the EDTA structure, implying a very strong binding/slow dissociation kinetics at these sites. In contrast, the closed sample also containing 2 Mg²⁺ (after desalting for ICP-MS) is consistent with the regulatory sites M3/M4 (are M1/M2 also involved in regulation as E374A/E378A does not affect proliferation?) being lower affinity/fast dissociating, which coincides with ~0.5 mM of matrix Mg²⁺ concentration. However, this also indicates ICP-MS is not suitable for evaluating Mg²⁺ binding at these lower affinity sites. Binding (e.g. ITC) and functional (e.g. electrophysiology) experiments comparing WT and mutants at these sites will be helpful to evaluate whether these sites are indeed functionally relevant.

Minor points:

1. Fig. 1e. Specific scale of electrostatic surface coloring (e.g. kT/e) would be more informative than "positive" and "negative".
2. Fig. 2 legends: is panel 'g' mis-labeled as 'c'?
3. Fig. 2e, f. It seems M3/M4 contains more negatively charged residues in hsMrs2 than in CtMrs2. Does this indicate difference in Mg²⁺ affinities?
4. line 250 does "...M4 is maintained in hMrs2..." mean "...M4 is maintained in hMrs2..."

Version 1:

Decision Letter:

Our ref: NSMB-A48236A

29th May 2024

Dear Dr. Gourdon,

Thank you for submitting your revised manuscript "Closed and open structures of the eukaryotic magnesium channel Mrs2 reveal the auto-ligand-gating regulation mechanism" (NSMB-A48236A). It has now been seen by the original referees and their comments are below. The reviewers find that the paper has improved in revision, and therefore we'll be happy in principle to publish it in Nature Structural & Molecular Biology, pending minor revisions to satisfy the referees' final requests and to comply with our editorial and formatting guidelines.

To facilitate our work at this stage, it is important that we have a copy of the main text as a word file. If you could please send along a word version of this file as soon as possible, we would greatly appreciate it; please make sure to copy the NSMB account (cc'ed above).

Sincerely,

Katarzyna Ciazynska, PhD
(she/her)
Associate Editor
Nature Structural & Molecular Biology
<https://orcid.org/0000-0002-9899-2428>

Reviewer #1 (Remarks to the Author):

The authors have adequately addressed the concerns. In particular, the additional functional data strongly support their conclusion. The authors have elegantly solved the long-standing mystery of channel gating in this family. I have only one minor point that needs to be corrected before publication.

1. Lines 505-506 "while CorB/CNNM transporters recognize fully dehydrated Mg²⁺ in the transmembrane domain 65."

The paper cited by the authors (Ref 65, PMID: 34188059) states that the Mg²⁺ ion in the CorB structure was modeled based on the previously published CorC TM domain structure (PMID: 33568487), which belongs to the same family. The Mg²⁺ bound structure of CorC was published earlier than the CorB structures and originally showed the recognition of the fully dehydrated Mg²⁺ in this family. Therefore, to be fair, the authors should also cite the CorC structure paper. In addition, the bacterial members of this family are generally referred to as both CorB and CorC. Likewise, the eukaryotic members of this family can be referred to as both CNNM and ACDP. Therefore, I would suggest that the authors rephrase "CorB/CNNM" as "CorB/CorC/CNNM/ACDP" in this sentence.

Reviewer #2 (Remarks to the Author):

The authors addressed all questions/concerns very well. This work represent important progress and should be published.

Version 2:

Decision Letter:

28th Oct 2024

Dear Dr. Gourdon,

We are now happy to accept your revised paper "Closed and open structures of the eukaryotic magnesium channel Mrs2 reveal the auto-ligand-gating regulation mechanism" for publication as an Article in Nature Structural & Molecular Biology.

Your paper will be published online soon after we receive proof corrections and will appear in print in the next available issue. You can find out your date of online publication by contacting the production team shortly after sending your proof corrections.

Please note that *Nature Structural & Molecular Biology* is a Transformative Journal (TJ). Authors may publish their research with us through the traditional subscription access route or make their paper immediately open access through payment of an article-processing charge (APC). Authors will not be required to make a final decision about access to their article until it has been accepted. [Find out more about Transformative Journals](https://www.springernature.com/gp/open-research/transformative-journals)

Sincerely,

Katarzyna Ciazynska, PhD
(she/her)
Senior Editor
Nature Structural & Molecular Biology
<https://orcid.org/0000-0002-9899-2428>

Response to reviewers

We thank the reviewers for the evaluation and for the helpful suggestions and comments to improve the manuscript "Closed and open structures of the eukaryotic magnesium channel Mrs2 reveal the auto-ligand-gating regulation mechanism". We have now addressed all comments and amended the manuscript as outlined below. Remarks and questions from the reviewers are shown in black. Our responses are shown in blue.

Reviewers' Comments:

Reviewer #1:

Remarks to the Author:

In this work, the authors determined the cryo-EM structures of *Chaetomium thermophilum* Mrs2 in the presence and absence of Mg²⁺ and performed the structure-based mutational analysis. Importantly, the structure in the Mg²⁺-free condition seems to represent the open conformation, which has not been reported in previous structural studies of Mrs2. Furthermore, the structural comparison of two structures revealed the symmetrical conformational changes, which is different from those observed in the prokaryotic CorA protein. The manuscript is well written and the data presented are solid. Overall, I strongly recommend this manuscript for publication if the following concerns are adequately addressed.

We thank reviewer for kind assessment of our manuscript.

Major concerns

1. Additional experiments to characterize the regulatory role of M1-M4 sites
The authors should verify the regulatory role of M1-M4 sites. The growth assay in Figure 2g is not appropriate to characterize this, as loss of the regulatory Mg²⁺ binding site would result in upregulation of Mg²⁺ uptake rather than loss of Mg²⁺ uptake. For example, the authors may try the Co²⁺ or Ni²⁺ sensitivity assay. Or the authors can also try limited proteolysis with purified CtMrs2 and its mutants of the M1-M4 sites in the presence and absence of Mg²⁺, as previously performed with TmCorA (PMID: 16902408). This is important because there is no previous functional report on CtMrs2. It is not clear even whether CtMrs2 is also an auto-ligand-gated ion channel like TmCorA. To properly interpret the Mg²⁺-dependent structural changes of CmMrs2 by cryo-EM, such functional experiments would be required.

Thank you for the informative suggestions to improve the manuscript. We have now employed four separate, additional, assays to further illuminate the role of the M1-M4 sites. As proposed by the reviewer, we set up a Ni²⁺-sensitivity assay, and we also conducted a limited proteolysis assay with CtMrs2 wild-type and mutants of the M1-M4 sites. In addition, we have also executed electrophysiological and ITC measurements as suggested by Reviewer #2. The results are shown in Fig. 5 in the revised manuscript. The results from all assays clearly support that the M1-M4 sites play an important role for the reversible transition between a closed and open channel, and for auto-regulation.

2. Addition of discussion section

In the current manuscript, the section "Conductance and gating mechanisms in eukaryotic Mrs2 proteins" seems to correspond to the discussion section, while the discussion section is missing from the manuscript. The authors should expand this section as a discussion section. For example, it would be interesting to discuss the

similarities and differences of the Mg²⁺-dependent regulation mechanism of Mrs2/CorA with that of other Mg²⁺ channels, which would further deepen the authors' findings.

We thank for reviewer for this suggestion, and we have now introduced a discussion about the Mg²⁺-dependent regulation mechanism of Mrs2/CorA family, and other Mg²⁺-channels and transporters in revised version.

3. Reason for the structural difference with the recent human Mrs2 structures

As the authors would be aware, very recently three independent groups reported the human Mrs2 structures in the presence and absence of Mg²⁺ ions (doi: <https://doi.org/10.1101/2023.08.22.553867>, doi: <https://doi.org/10.1101/2023.08.12.553106>, doi: <https://doi.org/10.1038/s41467-023-40516-2>). Interestingly, none of these structures showed the Mg²⁺-dependent structural changes at the conserved arginine (R413 in CtMrs2 and R332 in hMrs2) or the symmetric separation of the cytoplasmic domain. The authors should provide a reasonable explanation for this discrepancy. Is it because such structural changes are specific to CtMrs2 and not conserved in human Mrs2? If so, it would affect the significance of this work. Or does it reflect the different sample preparation conditions? All hMrs2 structures were determined in detergent, whereas CtMrs2 structures were determined in nanodisc. In addition, for hMrs2 structures in the presence of EDTA, EDTA was not added at the beginning of the purification for sample preparation.

This is very interesting question from the reviewer, and we are of course aware of the recently reported hMrs2 structures from three independent groups, and the fact that all of these hMrs2 structures represent the closed conformation, despite that some were obtained in the absence of Mg²⁺. We believe the open conformation shown in our work is also present in hMrs2. First, the ion conduct pathways and pore gating are highly similar between hMrs2 and CtMrs2, which suggest Mrs2 proteins share ion conduction mechanism. Secondly, it would be more energetically unfavourable for Mg²⁺ to cross the Met/Arg pore gate in hMrs2 without structural changes. Thirdly, the RDLR-motif is conserved among Mrs2 proteins, but not CorA proteins, suggesting the open conformation is stabilized in a similar manner in Mrs2 proteins, interacting with residues that form the M1 site in the closed conformation. Due to the missing RDLR-motif, the open conformation of CorA may look different.

It is likely that the different sample preparation procedures are responsible for the different conformations obtained in Mg²⁺-deficient conditions. We note we supplemented EDTA from the beginning of the purification, even before solubilization when the protein was still in the native membranes, for generation of the open structure. In contrast, the hMrs2 structures generated in the presence of low Mg²⁺ were obtained from samples treated with EDTA when the protein was already in detergent or even only immediately prior to the structural analysis. It is also likely that the nano-disc environment further stabilizes the open configuration retrieved by the EDTA treatment. Indeed, the lipidic environment is critical for the function of membrane protein (see e.g. PMID: 23451886). Since our nano-discs represent a better membrane mimic, it is possible this lipidic environment assisted capturing the open CtMrs2 state.

Minor concerns

1. Supplementary Fig. 2.

Please include CorA proteins in the sequence alignment. It is important to show that Arg413 is not conserved in CorA.

Certain relevant CorA proteins have been included in the sequence alignment, showing that Arg413 is conserved in Mrs2 family, but not in CorA members.

2. Page 9. "Similarly, the G441A/M442A/N443A form exhibited reduced cell proliferation, which is consistent with a previous study demonstrating interruptions of the GMN-motif impair the protein function."

Please cite the relevant papers.

A few different relevant citations have now been introduced to support the statement.

3. Fig. 2g

There is a typo. N14A should be corrected to N414A.

Corrected.

4. Legend of Fig. 2d. "Site U is positioned next to the loop in-between TM1 and TM2 and site S at the GMN-motif if the selective filter".

The part "if the selectivity filter" might be a typo? It should be corrected to "of the selectivity filter"?

Corrected.

5. Legend of Fig. 2d. "Sites T1 and T2 are located in the transmembrane domain. e, Close-views of sites M1-M4 in the N-terminal domain of CtMrs2".

T1 and T2 would be a typo and should be corrected as P1 and P2.

Corrected.

6. Fig. 3.

The authors should also illustrate the structural difference of R406, as it is the most constricted site in the closed state.

The figure has been revised to illustrate the structural difference of R406 between the closed and open states.

7. Fig. 4c, d.

The way the authors show M1/M2 (cartoon) and M3/4 (surface) is not consistent. Please show M1/M2 and M3/M4 in both cartoon and surface representations.

We now show M1/M2 and M3/M4 in both cartoon and surface representations in the revised version.

8. Supplementary Fig. 5a.

How did the authors obtain the AlphaFold model of ScMrs2? By making the model themselves? Or by downloading it from a database? Please clarify, and if they generated the model themselves, please provide the model and log files of AlphaFold.

The AlphaFold model of ScMrs2 for structural comparison was downloaded from AlphaFold Protein Structure Database (via UNIPROT). This information is now provided in the manuscript.

Reviewer #2:

Remarks to the Author:

CorA/Mrs2 proteins play essential roles in cation permeation across cellular membranes, especially Mg²⁺ import into mitochondria matrix. How these channels work remained unclear in a large part due to the absence of conductive/open conformation structure. The authors present here near-atomic resolution structures of a eukaryote (fungus) Mrs2 protein in conformations consistent with both closed and open states in lipid nano-discs. The Mg²⁺ autoregulation site has been proposed. Overall, this work provides important new open structural information that promotes the understanding this family of proteins. Evaluation of functional relevance in the proposed Mg²⁺ regulation sites would further test the proposed working mechanism. With a few other minor clarifications/edits, this work should be shared with the community.

We thank the reviewer for the careful evaluation and positive assessment of our manuscript.

Major point:

1. Mg²⁺ binding and autoregulation. The same Mg²⁺:Mrs2 stoichiometry of ~2 across EDTA-treated/M3-M4 mutants/Mg²⁺ samples is interesting and also raises some concerns. Mg²⁺ was built in the S, P1 and P2 sites even in the EDTA structure, implying a very strong binding/slow dissociation kinetics at these sites. In contrast, the closed sample also containing 2 Mg²⁺ (after desalting for ICP-MS) is consistent with the regulatory sites M3/M4 (are M1/M2 also involved in regulation as E374A/E378A does not affect proliferation?) being lower affinity/fast dissociating, which coincides with ~0.5 mM of matrix Mg²⁺ concentration. However, this also indicates ICP-MS is not suitable for evaluating Mg²⁺ binding at these lower affinity sites. Binding (e.g. ITC) and functional (e.g. electrophysiology) experiments comparing WT and mutants at these sites will be helpful to evaluate whether these sites are indeed functionally relevant.

It is correct that we believe that S, P1 and P2 sites remain occupied also in the presence of EDTA. This is consistent with the recently reported hMrs2 structures, where Mg²⁺ binds to the S and the P2 sites (PDB-ID: 8TUP) under Mg²⁺-free conditions, and to the S site under low EDTA (1 mM, PDB-ID: 8IP4) and high EDTA (5 mM, PDB-ID: 8IP5) conditions. We agree with the reviewer that this is implying a strong binding/slow dissociation kinetics at these sites, specifically for the S site (at the GMN-motif), which is a defining feature for the CorA/Mrs2 protein family. We note the CorA Mg²⁺ affinity has been estimated to $1.3 \pm 0.1 \mu\text{M}$, PMID: 24516146). Such Mg²⁺ occupation of S site may help stabilizing the pore, and prevent (back)flow/binding of other Mg²⁺/ions.

We have now employed four separate, additional, assays to further illuminate the role of the M1-M4 sites. As proposed by the reviewer, we have executed electrophysiological and ITC measurements, comparing CtMrs2 wild-type and mutants of the M1-M4 sites. We have also setup a Ni²⁺-sensitivity assay, and conducted a limited proteolysis assay, as proposed by reviewer#1. The results are shown in Fig. 5 in the revised manuscript. The results from all assays clearly support that the M1-M4 sites play an important role for the reversible transition between a closed and open channel, and for auto-regulation, as Mg²⁺ sensors in the mitochondrial matrix. Evaluation of the binding to the M1-M4 low-affinity sites is shown in Fig. 5f and Supplementary Fig. 13).

The ITC data are consistent with a sequential binding model with high cooperativity between binding to the M1/M2 sites and M3/M4 sites. This implies that binding to the M3/M4 sites only happens when Mg²⁺ is bound to the M1/M2 sites and the channel has undertaken a conformational change from the open to the closed state. The ITC data may be interpreted as binding to M3/M4 sites are associated with higher affinity than the

binding to the M1/M2 sites. However, accurate quantitative analysis of the data is difficult due to the complexity of the reactions and the limited amount of data points. Collectively, our structural and functional data point to a regulatory mechanism of the M1-M4 sites where the channel is sensitive to high concentrations of Mg^{2+} (in the mM-range) which leads to binding to M1/M2 and conformational changes that leads to the closed state. In the latter form, the M3/M4 sites are formed and Mg^{2+} binds, thereby stabilizing the conformation.

Minor points:

1. Fig. 1e. Specific scale of electrostatic surface coloring (e.g. kT/e) would be more informative than "positive" and "negative".

We have changed from "positive" and "negative" to "kT/e" in all figures where the electrostatic surface is shown.

2. Fig. 2 legends: is panel 'g' mis-labeled as 'c'

Corrected.

3. Fig. 2e, f. It seems M3/M4 contains more negatively charged residues in hMrs2 than in CtMrs2. Does this indicate difference in Mg^{2+} affinities?

Indeed, there are more negatively charged residues at sites M3/M4 in hMrs2 than in CtMrs2, and thus it is possible that those sites show higher affinity for Mg^{2+} in hMrs2 than in CtMrs2.

4. line 250 does "...M4 is maintained in hMrs2..." mean "...M4 is maintained in hMrs2..."

Corrected.

Response to reviewers

We thank the reviewers again for the evaluation of the manuscript "Closed and open structures of the eukaryotic magnesium channel Mrs2 reveal the auto-ligand-gating regulation mechanism". We have now addressed all comments and amended the manuscript as outlined below. Remarks and questions from the reviewers are shown in black. Our responses are shown in blue.

Reviewers' Comments:

Reviewer #1:

Remarks to the Author:

The authors have adequately addressed the concerns. In particular, the additional functional data strongly support their conclusion. The authors have elegantly solved the long-standing mystery of channel gating in this family. I have only one minor point that needs to be corrected before publication.

We thank the reviewer for the endorsement of our manuscript.

1. Lines 505-506 "while CorB/CNNM transporters recognize fully dehydrated Mg²⁺ in the transmembrane domain 65."

The paper cited by the authors (Ref 65, PMID: 34188059) states that the Mg²⁺ ion in the CorB structure was modeled based on the previously published CorC TM domain structure (PMID: 33568487), which belongs to the same family. The Mg²⁺ bound structure of CorC was published earlier than the CorB structures and originally showed the recognition of the fully dehydrated Mg²⁺ in this family. Therefore, to be fair, the authors should also cite the CorC structure paper.

We thank the review pinpointed this and we have now updated our citation with CorC structure paper.

In addition, the bacterial members of this family are generally referred to as both CorB and CorC. Likewise, the eukaryotic members of this family can be referred to as both CNNM and ACDP. Therefore, I would suggest that the authors rephrase "CorB/CNNM" as "CorB/CorC/CNNM/ACDP" in this sentence.

We thank the reviewer for this suggestion. We have now rephrased "CorB/CNNM" as "CorB/CorC/CNNM/ACDP" in this sentence. This part of the manuscript has been moved to the Supplementary discussion in the interest of saving space (words) as indication by the editor.

Reviewer #2:

Remarks to the Author:

The authors addressed all questions/concerns very well. This work represent important progress and should be published.

We thank the reviewer again for the evaluation and positive assessment of our manuscript.